# Construction of a mammalian embryo model from stem cells organized by a morphogen signalling centre

Peng-Fei Xu [1,3,5], Ricardo Moraes Borges [1,5], Jonathan Fillatre [1,5], Maraysa de Oliveira-Melo [1,4], Tao Cheng [2], Bernard Thisse[1] & Christine Thisse [1✉]

Generating properly differentiated embryonic structures in vitro from pluripotent stem cells remains a challenge. Here we show that instruction of aggregates of mouse embryonic stem cells with an experimentally engineered morphogen signalling centre, that functions as an organizer, results in the development of embryo-like entities (embryoids). In situ hybridization, immunolabelling, cell tracking and transcriptomic analyses show that these embryoids form the three germ layers through a gastrulation process and that they exhibit a wide range of developmental structures, highly similar to neurula-stage mouse embryos. Embryoids are organized around an axial chordamesoderm, with a dorsal neural plate that displays histological properties similar to the murine embryo neuroepithelium and that folds into a neural tube patterned antero-posteriorly from the posterior midbrain to the tip of the tail. Lateral to the chordamesoderm, embryoids display somitic and intermediate mesoderm, with beating cardiac tissue anteriorly and formation of a vasculature network. Ventrally, embryoids differentiate a primitive gut tube, which is patterned both antero-posteriorly and dorso-ventrally. Altogether, embryoids provide an in vitro model of mammalian embryo that displays extensive development of germ layer derivatives and that promises to be a powerful tool for in vitro studies and disease modelling.

[1] Department of Cell Biology, University of Virginia, Charlottesville, VA, USA. [2] Institute of Genetics and Department of Genetics, School of Medicine, Zhejiang University, Hangzhou, China. [3] Present address: Institute of Genetics and Department of Genetics, School of Medicine, Zhejiang University, Hangzhou, China. [4] Present address: Department of Cell Biology, State University of Campinas, Campinas, Brazil. [5] These authors contributed equally: Peng-Fei Xu, Ricardo Moraes Borges, Jonathan Fillatre. ✉email: christhisse@virginia.edu

In recent years, there has been substantial effort to reproduce mammalian embryonic development in vitro. First, multi-cellular structures mimicking part of organs known as organoids[1,2] have been generated using stem cells (either embryonic stem cells (ESC) or induced pluripotent stem cells (iPSCs)) grown within a basement membrane gel and that develop into 3D shapes and self-organize in response to the presence in the culture medium of signalling molecules, such as morphogens or morphogen antagonists. While organoids are able to mimic microanatomy of authentic organs, they display minimal organization and lack the full variety of cell types that are derived from the three germ layers. To achieve a much higher degree of organization, organs need to be vascularized and innervated and contain the full spectrum of cells and tissues type derived from the three germ layers that can be patterned and organized along the three main body axes.

Toward this goal, recent experimental strategies have allowed the development of in vitro models of embryos[3,4]. Two main strategies have been explored. The first approach used a combination of stem cells from both embryonic and extra-embryonic origins. Combination of ESC and trophoblast stem cells (TSC) in a nonadherent platform led to the formation of pre-implantation embryo-like structures similar to blastocysts, called "blastoids"[5,6]. However, blastoids failed to develop past pre-implantation stages. When placed in a 3D culture, the combination of ESC and TSC led to self-organized entities similar to an early post-implantation embryo but one that lacked visceral endoderm and failed to gastrulate[7]. Combining ESC and TSC with extra-embryonic endoderm stem cells (XEN), resulted in embryo-like structures that exhibited an epithelial–mesenchymal transition, leading to mesoderm and definitive endoderm specification, similar to an embryo at mid-gastrulation[8]. However, these structures failed to develop further. Therefore, taking advantages of the properties of embryonic and extra-embryonic stem cells allowed the formation of good morphological copies of gastrulating embryos but thus far they have not been able to develop beyond the gastrula stage to reach the vertebrate phylotypic stage (around embryonic day 8.0–8.5 in the mouse embryo[9]). The main limitation of using this strategy is probably related to the fact that it relies on self-organization of different stem cell strains, a process that is not subject to direct experimental controls. The second strategy relies on the self-organization properties of ESC aggregates exposed to a pulse of the WNT/β-catenin signalling agonist CHIR99021 after 2 days in culture[10–12]. The treated aggregates spontaneously formed a pole of brachyury (Bra) expression, they elongated, expressed genes found in a mouse embryo up to embryonic day 9.5 and displayed the typical bilateral symmetry of vertebrate embryos. These structures, named "gastruloids", mimicked embryonic development of the post-occipital region of the mouse embryo[11] but did not develop any cephalic structures. Moreover, despite specification of cell fates and some patterning of tissues[12], these gastruloids showed limited morphogenesis, lacked proper tissues organization and were devoid of a neural tube. More recent studies revealed that embedding ESC aggregates in Matrigel, led to the formation of gastruloids that developed somites, appearing sequentially with correct antero-posterior (AP) patterning[13]. An improved method involving addition of the WNT agonist together with a BMP antagonist resulted in formation of trunk-like structures (TLS)[14] that displayed somites and neural tube. In another report, a similar method was shown to support generation of cardiovascular progenitors that self-organized into an anterior domain reminiscent of a cardiac crescent before forming a beating cardiac tissue[15]. As well, it was reported that using synthetic matrices, in addition to cultivating ESC under neural induction conditions, led to the formation of neuroepithelial cysts[16–18].

Similar to blastuloids, gastruloids and TLS formation depend on self-organization of stem cell aggregates in response to cell culture conditions. Development into gastruloids/TLS was achieved by activating the WNT/β-catenin signalling pathway within stem cell aggregates resulting in the formation of a posterior pole expressing the mesodermal marker Bra. Because all cells of the aggregates were exposed to the WNT agonist, gastruloids/TLS lacked the most anterior structures and resembled Dkk mutants for which anterior domains are posteriorized due to the excess of WNT signalling[19].

One limitation that may explain why only partial development is obtained in vitro may be related to the method used to induce developmental programmes in ESC. In vertebrates, the signals breaking the initial spherical symmetry of the egg and instructing cells about their fate and behaviour, are present in the egg and/or are generated by cells of the embryo itself or by its extra-embryonic tissues (primitive streak, extra-embryonic ectoderm and visceral endoderm in the mouse embryo). Therefore, the developmental programmes that control patterning and morphogenesis are not induced by soluble signals present in the liquid medium surrounding the embryo. Conversely, induction of the development programmes in gastruloids or TLS results from the incubation of ESC aggregates in a culture medium containing a small molecule agonist of the WNT/β-catenin signalling pathway, which is homogenously distributed around the ESC aggregates. By using this protocol, all superficial cells of the aggregates are exposed to the same level of stimulation, which depends solely on the concentration of the WNT agonist in the medium. The mechanism resulting into the breaking of symmetry of gastruloids and TLS is not yet understood but this may involve a cell sorting of mesodermal cells, that may derive from the superficial cells being exposed to a higher concentration of agonist while deeper cells may be less stimulated and elicit an ectodermal fate. Despite this breaking of symmetry, it is likely that all cells of the aggregate have been exposed to a level of WNT agonist and hence a degree of activation of the WNT pathway known to promote posterior identities. This may explain why gastruloids and TLS can only generate post-occipital structures.

Therefore, being able to experimentally control in vitro the formation of extensive and properly organized embryo-like structures requires designing spatially restricted signalling centre(s). To do so, attempts have been made at generating localized signals using microfluidic technologies[20,21]. However, thus far, no extensive embryonic development with differentiation of derivatives from all three germ layers has been reported using this strategy.

Our lab has previously shown that experimentally generated opposing gradients of BMP and Nodal (that induces expression of zygotic Wnt) is sufficient to induce the formation of a complete zebrafish embryonic axis in vitro, either using naive pluripotent cells of animal pole explants[22,23] or using aggregates of dissociated and reassociated animal pole blastomeres[24]. Based on these findings, we hypothesized that extensive embryonic development could be achieved by engineering a localized morphogen signalling centre within an aggregate of mouse ESC.

Here we show that embryo-like entities (we name 'embryoids') are able to be generated by instructing an aggregate made of naive (untreated) ESC merged with another aggregate, in which expression of WNT3 and NODAL has been induced. Cells of the instructing aggregate function like a signalling centre that controls patterning and morphogenesis of the entire structure leading to the formation of tissues and organs. The signal provided by the instructed cells triggers the expression of downstream factors, both eliciting and executing the developmental programmes encoded in the genome, mimicking mouse embryonic development. Indeed, the embryoids elongate, form the three germ layers

through a process of gastrulation and differentiate germ layer derivatives. Embryoids are patterned along their AP and dorsal-ventral (DV) axes and display extensive features similar to a neurula-stage mouse embryo.

## Results

**Engineering a morphogen signalling centre.** In the mouse embryo, the initial organization of the gastrula results from the interplay of three signalling molecules, the morphogens BMP4, NODAL and WNT. First, the secretion of BMP4 by the extra-embryonic ectoderm promotes expression of WNT and NODAL in the embryonic epiblast[25]. Antagonists of these morphogens (DKK1, CER-1 and LEFTY1) secreted by the anterior visceral endoderm restrict their expression and activity to the posterior domain of the embryo where they induce the formation of the primitive streak, a structure in which epiblast cells undergo an epithelial-to-mesenchymal transition giving rise to mesoderm and endoderm[26]. With the goal to reproduce in vitro the activity of the morphogens present in the mouse embryo at early gastrula stage, we engineered a signalling centre secreting WNT3 and NODAL within aggregates of mouse ESC. To do so, aggregates of two different sizes (50 and 100 cells) were generated at Day 0 (D0), then cultured for 3 days in a basal medium devoid of factors maintaining cell pluripotency. On the third day (D3), small aggregates (50 cells) were incubated for 8 h in the presence of purified mouse BMP4 protein. As expected, we observed that incubation of the ESC aggregates with BMP4 resulted in inducing expression of WNT3 and NODAL that in turn induced expression of their downstream target genes such as *Eomesodermin* (*Eomes*) and *Brachyury* (*Bra*) (Supplementary Fig. 1). These BMP4 treated (instructed) aggregates were then individually placed at the bottom of wells of ultra-low attachment plates in contact with individual large aggregates, made of 100 cells and that were not exposed to BMP4 stimulation (untreated aggregates). Within 1 h, instructed and untreated aggregates merged spontaneously (Fig. 1a–c) ($N = 5154/5184$ successful merging, scored in 11 experiments, giving a 99.29% success rate; Supplementary Table 1) into a larger structure thereafter called an "embryoid", in which the instructed aggregate served as an organizing centre.

We confirmed by in situ hybridization that activation by BMP4 led to expression of WNT3 and NODAL in the signalling centre, which were asymmetrically located at one tip of the growing embryoids (Fig. 1d, e). Secretion of these morphogens resulted in inducing expression of the mesoderm and primitive streak marker *Bra* within the signalling centre (Fig. 1f). However, *Bra* expression was not restricted uniquely to cells of the signalling centre but was also observed induced in an adjacent stripe of cells from the untreated aggregates (Fig. 1g–i). This showed that NODAL and WNT3 promoted mesodermal identity in cells of both instructed and untreated aggregates. At D4, *Bra* expressing cells were observed clustered posteriorly while *Otx2* transcripts were detected in the entire embryoids as observed in the mouse embryo at E6.0[27] (Fig. 1j). A day later (D5), similar to its expression in the mouse embryo, *Otx2* transcripts became restricted to the anterior half of the embryoids (Fig. 1k). At D5.5, embryoids began to elongate (Supplementary Fig. 2) and to bend into a bean shape (Fig. 1k–m). The variability in shape amongst the population of embryoids is presented in Supplementary Fig. 3. Neural progenitors, fluorescently labelled in embryoids made of *Sox1-GFP* ESC[28] (expressing *GFP* under the control of the *Sox1* promoter) in both the naive territory and in the signalling centre, were found at the anterior end of the embryoids. *GFP* expression was also seen on the convex side of the embryoids, identifying their dorsal side (Fig. 1l).

The mesoderm, *GFP* positive in embryoids made of *Bra-GFP* ESC for both the naive aggregate and the signalling centre, occupied a complementary territory to *Sox1*, which was located in ventral and posterior regions (Fig. 1m). At D6, in situ labelling for *Chrd* revealed a notochord-like structure present on the dorsal side of the embryoids, providing further evidence for DV patterning (Fig. 1n). At D7, a large cavity, surrounded by an epithelium that may correspond to the endoderm epithelium, was located ventrally to this notochord-like structure (Fig. 1o). The succession of neural tissue, axial mesoderm and presumptive endoderm, from the convex to the concave sides of the embryoids strongly supported the presence of a DV axis in the growing embryoids. We also found clear evidence of an AP axis (Fig. 1p). At D6.5, one colour in situ hybridization for *Wnt3a* labelled the posterior end of the embryoids. Double colour in situ hybridization for *Otx2* and *Bra* revealed an anterior *Otx2* expressing domain and a posterior *Bra* expressing domain (likely equivalent to the primitive streak of the mouse embryo) that were separated by a large median domain in which a notochord-like structure was visible on top of the presumptive endoderm epithelium surrounding the central cavity (Fig. 1q).

**Gastrulation-like process in the embryoids.** At early stages of development, *Bra* expressing mesodermal cells present in the posterior side of the embryoids displayed a mesenchymal organization (Supplementary Fig. 5). In between D3.5 and D5, they progressively moved anteriorly in a process reminiscent of the anterior-ward migration of mesodermal cells during gastrulation of a mammalian embryo. To compare the migratory behaviour of mesodermal cells in embryoids to what has been reported for mouse embryos, we used two independent experimental conditions in which individual mesodermal cells were labelled and their position along the embryoid axis recorded. In the first experiment (Fig. 2a–c) embryoids were prepared by merging at D3 a naive aggregate made of unlabelled ESC with a BMP4 instructed aggregate made at D0 from 90% (45 cells) of unlabelled ESC and 10% (5 cells) of *Bra-GFP* ESC (reducing the number of *Bra-GFP* cells allows for the precise determination of the position of individual *Bra* expressing cells) (Fig. 2a). As described previously, BMP4 induction of WNT3 and NODAL expression in the signalling centre resulted in the expression of *Bra* in all mesodermal cells and the expression of *GFP* in the 10% *Bra-GFP* ESC present in the signalling centre. *Bra-GFP* cells initially restricted to the signalling centre (D3.5) were progressively found in more anterior positions (D4, D4.5), strongly suggesting that they have migrated from the primitive streak-like domain, located at one end (likely the posterior end) of the ovoid-shaped embryoids, towards the opposite end (likely the anterior end) (Fig. 2b, c). A control experiment using 100% *Bra-GFP* cells in the signalling centre led to the same observations even though we were unable to track them individually (Supplementary Fig. 5).

In a second experiment (Fig. 2d–f), naive aggregates made at D0 with 90 unlabelled ESC and 10 *Bra-GFP* ESC were merged at D3 with BMP4 instructed aggregates composed only of unlabelled ESC (Fig. 2d). In the resulting embryoids, (shown in Fig. 1i), WNT3 and NODAL secreted by the unlabelled cells of the signalling centre induced mesoderm and therefore expression of *GFP* in *Bra-GFP* cells present in the naive part of the embryoid in the domain adjacent to the signalling centre. Conversely to our first experiment (Fig. 2a–c) where *GFP* labelled cells from the signalling centre at the posterior end were observed progressively into more anterior position, at D4, *GFP* expressing cells induced in the naive territory were found in a more posterior position than at D3.5, supporting the conclusion that the first movement of these cells was toward the posterior end of the embryoids.

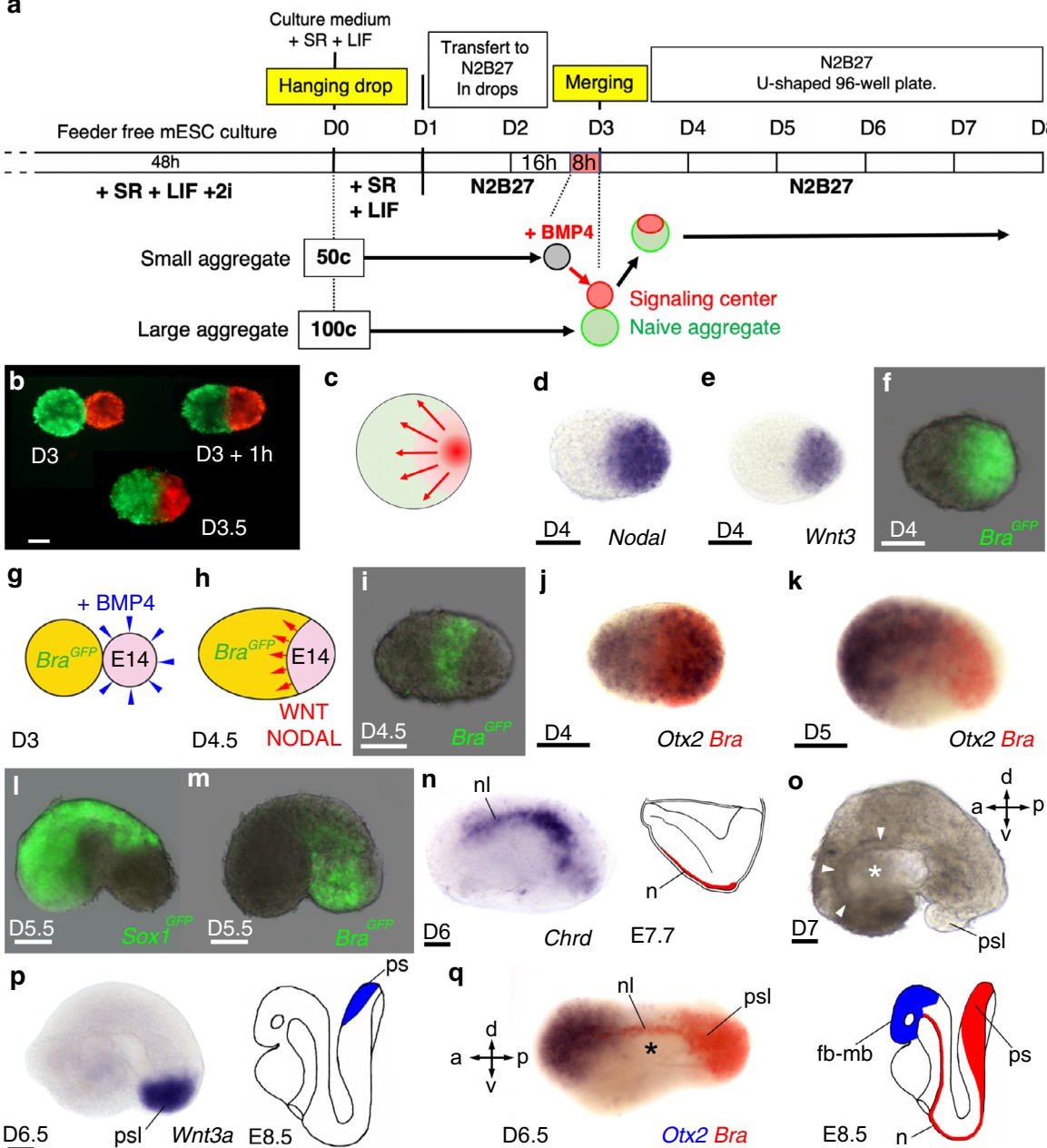

**Fig. 1 Formation of embryoids by instruction of ESC aggregates with a morphogen signalling centre. a** Experimental procedure (detailed in 'Methods') to generate embryoids by merging a large naive (untreated) ESC aggregate with a smaller aggregate in which expression of WNT3 and NODAL morphogens were induced following incubation with BMP4 protein. **b** A naive ESC aggregate labelled with DiO (green) placed in contact at D3 with a BMP4 instructed ESC aggregate labelled with DiI (red) had merged within 1 h and formed a spherical embryoid after 12 h. (D3.5). **c** Schematic summarizing the principle of instructing an ESC aggregate with a morphogen signalling centre. **d**, **e** In situ hybridization at D4 for **d** *Nodal* and **e** *Wnt3*. **f** Expression at D4 of *Bra*GFP in an embryoid made of *Bra-GFP* ESC. **g**, **h** Experimental design to visualize the induction of *Bra* expression (**i**) in *Bra-GFP* ESC (naive aggregate, yellow) in response to morphogens secreted by the signalling centre (pink) made of unlabelled E14TG2 ESC. **j**, **k** Double colour in situ hybridization for *Otx2* (blue) and *Bra* (red). **l**, **m** Expression of *Sox1*GFP identifying the neurectoderm (**l**) and of *Bra*GFP identifying the mesoderm (**m**). **n** Expression of *Chrd* (nl: notochord-like) in a D6 embryoid, right: drawing for *Chrd* expression in mouse embryo at E7.7. **o** D7 embryoid showing a large cavity (star) lined by an epithelium (arrows). **p** Expression at D6.5 of *Wnt3a* in a posterior primitive streak-like structure (psl) and in mouse at E8.5 (cartoon). **q** Double colour in situ hybridization showing expression of *Otx2* (blue) anteriorly, of *Bra* (red) posteriorly and in notochord-like structure extending on top of a ventral cavity (star) in an embryoid at D6.5. Cartoon showing expression of *Bra* and *Otx2* in the mouse embryo at E8.5 (fb-mb: forebrain-midbrain, n: notochord, ps: primitive streak). a: anterior, p: posterior, d: dorsal, v: ventral. Staging of embryoids (D: day) indicated for each panel. Numbers of performed experiments and of biological samples analysed is provided in 'Statistics and reproducibility' in the Methods section. Variability in genes expression is provided in Supplementary Fig. 4. Scale bars: 100 μm.

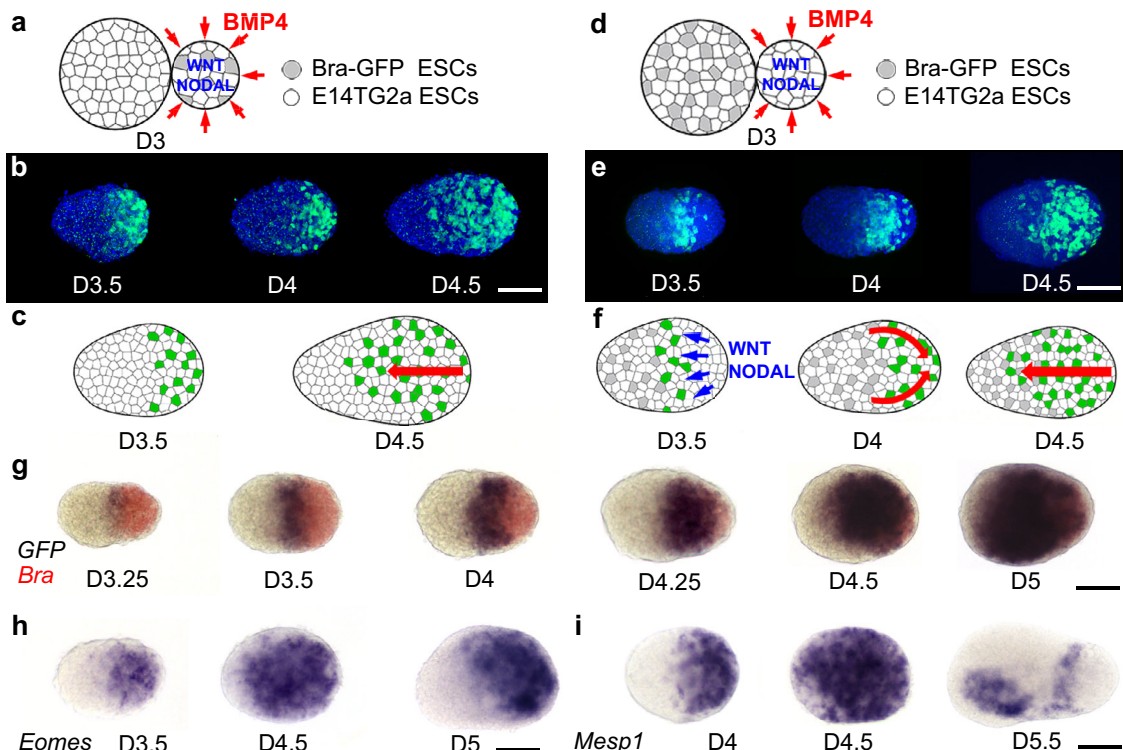

**Fig. 2 Position over time of mesodermal cells during the gastrulation of embryoids. a–c** Position over time of mesodermal cells induced in the signalling centre composed of 90% of unlabelled cells and 10% of *Bra-GFP* ESC. **a** Schematic of the experiment. **b** Visualization of *Bra*^GFP expressing cells (green) and of cell nuclei (Hoechst—blue) in fixed embryoids (D3.5 to D4.5). **c** Interpretive drawing: mesodermal cells induced by WNT3 and NODAL in the signalling centre were observed progressively more anteriorly during gastrulation (red arrow). **d–f** Position over time of mesodermal cells induced in the naive aggregate made of 90% unlabelled cells and 10% *Bra-GFP* ESC by WNT3 and NODAL secreted by the signalling centre. **d** Schematic of the experiment. **e** Visualization of *Bra*^GFP (green) and of cell nuclei (Hoechst—blue) in fixed embryoids from D3.5 to D4.5. **f** Interpretive drawing (red arrow: change in position of *Bra*^GFP). WNT3 and NODAL secreted by unlabelled cells of the signalling centre induced expression of the *Bra*^GFP in a stripe of adjacent cells deriving from the naive aggregate. At D4, *Bra*^GFP cells were found close to the posterior end of the embryoids and at D4.5, cells were also observed in the anterior half of the embryoids. **g** Double colour in situ hybridization for *Bra* (red—all mesodermal cells) and *GFP* (blue—*Bra-GFP* mesodermal cells induced in the naive territory) from D3.25 to D5 with the same experimental design as in (**d**). Cells induced in the naive territory (expressing *Bra*^GFP) were observed in progressively more posterior territories (D3.25–D4.25), then from D4.5 to D5, were also observed in the anterior domain of the embryoids. **h, i** In situ hybridization for (**h**) *Eomes* and (**i**) *Mesp1*. **h** *Eomes* expressing cells, initially expressed posteriorly (D3.5), were then observed throughout the embryoids (D4.5), then expression disappeared in their anterior domain and was maintained in their posterior and dorsal sides (D5.0). **i** *Mesp1* expression was initially observed posteriorly (D4) and later throughout the entire embryoids (D4.5). It then disappeared anteriorly (D5.5) except in the ventral side (likely heart field) and appeared in a stripe located in the anterior presomitic mesoderm. Embryoids oriented anterior left, dorsal up. Scale bars: 100 μm. Numbers of performed experiments and of biological samples analysed is provided in 'Statistics and reproducibility' in the Methods section. Variability in genes expression is provided in Supplementary Fig. 7.

However, a day later (D4.5) these *Bra-GFP* labelled cells were found also in the anterior half of the embryoids (Fig. 2e, f). The positions along the AP axis of the *Bra-GFP* mesodermal cells corresponding to this experiment (Fig. 2d–f) were determined from D3.25 to D5 and strongly supports the conclusion that there was an initial move of these cells toward the posterior pole before they initiated a movement toward the anterior part of the embryoids (Supplementary Fig. 6).

Localization over time of mesodermal cells was further studied using double colour in situ hybridization at the same developmental stages (D3.25–D5) in embryoids made by merging naive aggregates made of *Bra-GFP* ESC with BMP4 instructed aggregates made of unlabelled ESC. We compared the expression of *Bra* (which labels all mesodermal cells both in the signalling centre and in cells of the untreated aggregates) with that of *GFP* (which labels *Bra-GFP* expressing cells induced by the signalling centre in cells of the naive aggregates). We found that *Bra-GFP* expressing cells, revealed by the *GFP* RNA probe, formed an

initial stripe located at around 40% from the posterior end of the embryoids. At D4–D4.25, *GFP* RNA containing cells were observed in a progressively more posterior position before they started to be detected (D4.5–D5) in the anterior part of the embryoids (Fig. 2g), supporting the migratory behaviour observed by tracking the fluorescence of *Bra-GFP* labelled cells. These data further confirmed that the anterior cells expressing *Bra-GFP* at the onset of gastrulation first moved posteriorly, then after D4.5, moved anteriorly. Altogether, the migratory behaviour of these mesodermal cells is reminiscent of that of a mouse embryo at the onset of gastrulation in which the first cells migrating away from the primitive streak are the most posterior cells, which then are replaced at their posteriormost position by their anterior neighbours. This is similar to the behaviour of mesodermal cells in other gastrulating vertebrates such as fish and frog for which the first cells involuting at gastrulation are the most vegetal cells of the embryonic margin while the mesodermal cells located further from the margin, first move vegetally before

involuting, reverse direction, moving in the opposite direction toward the animal pole of the embryo.

The posterior to anterior movement of the mesodermal cells during embryoid gastrulation was further confirmed by examining at various time points (D3.25–D4.5) the expression of *Eomes* and *Mesp1* (Fig. 2h, i), two genes specific for migrating mesodermal cells in the mesodermal wings of the mouse embryo[29]. In mouse, *Mesp1* is known to be induced by *Eomes*[30], therefore, its expression is delayed for about half a day. This was also what we observed in the embryoids. By D3.5 for *Eomes* and D4 for *Mesp1*, the mesodermal cells expressing these genes were located posteriorly, then they were progressively observed in more anterior positions until they reached the anterior tip of the embryoids by D4.5 (Fig. 2h, i). Expression of both *Eomes* and *Mesp1* was then progressively excluded from the anterior part of the embryoids. In the mouse embryo, *Eomes* becomes progressively restricted to the anterior primitive streak and to the mesoderm that is most proximal to the streak[31]. Similarly, in the embryoids at D5, transcripts of *Eomes* have disappeared from the anterior territory and were found posteriorly as well as on the dorsal side (Fig. 2h), in a domain likely equivalent to the anterior primitive streak. For *Mesp1*, expression disappeared completely in the dorsal anterior domain but was still observed antero-ventrally (Fig. 2i). In the mouse embryo, *Mesp1* is described to be an early marker for heart precursors. Therefore, the ventral anterior domain of *Mesp1* expression in the embryoids likely corresponds to the heart field[32]. In the mouse embryo, *Mesp1* quickly disappears from the anterior territories at the end of gastrulation and starts to be expressed in the anterior presomitic mesoderm (PSM)[33]. This anterior PSM expression domain was present at D5 in the embryoids but concomitant with anterior expression of *Mesp1*, suggesting that a mechanism negatively regulating *Mesp1* expression anteriorly at the end of gastrulation may be absent in the embryoids.

In summary, embryoids show clear evidence of a gastrulation process involving mesodermal cell movements that appear similar to those observed in the mouse embryo, with a primitive streak-like domain located at the posterior tip from where cells migrate anteriorly to form the mesoderm germ layer.

**Formation of endodermal derivatives.** The position of neuroectodermal cells expressing *Sox1-GFP* in an anterior dorsal position and the localization of mesodermal cells expressing *Bra-GFP* in the complementary ventral posterior domain of the embryoids (Fig. 1l, m) strongly support the conclusion that different germ layers were established during early stages of embryoid development. We looked for evidence of endoderm, including formation of a primitive endoderm (PrE) cell population, a visceral endoderm (VE) epithelium, its invasion by definitive endoderm (DE) cells and formation of the primitive gut.

In the mouse embryo, PrE appears at blastula stage within the inner cell mass, which is composed of two cell populations: a population of epiblast cells expressing NANOG and a population of PrE cells expressing GATA6[34]. With our culture condition (neurobasal medium—N2B27), untreated aggregates of ESC merged together at D3 showed weak expression of NANOG and only a few aggregates displayed sparse expression of GATA6 at D4 and D5 (Supplementary Fig. 8).

However, embryoids generated by merging a naive aggregate with a BMP4 activated aggregate displayed a "salt and pepper" expression pattern for NANOG and GATA6 at D3 and D4 (Fig. 3a, b). During this early phase of development (D3.5–4.5), GATA6(+) VE-like cells were sorted out from NANOG(+) cells (Fig. 3c), similar to the cell sorting occurring in the blastula stage mouse embryo and leading to PrE epithelium formation[35]. We

also labelled embryoids with Dolichos Biflorus Agglutinin (DBA) (VE marker[36,37]); this revealed the presence of a cloud of VE-like cells (Fig. 3d). Cluster of VE-like cells accumulated either at the surface of the embryoids (Fig. 3e, similar to PrE in the mouse embryo, which is located at the interface between the epiblast and the blastocoel cavity) or they formed clumps expressing *Gata6* (Fig. 3f) and *Sox17* (another endodermal marker expressed in PrE, VE and DE[38]) (Fig. 3g). Clumps of cells reaching the surface of the embryoids arranged themselves in a loosely organized epithelium that never completely covered the embryoids (Fig. 3g). Cells at the surface were randomly distributed and formed epithelial structures highly variable in size. We hypothesized that these VE-like cells at the outer surface of the embryoids may represent an equivalent, in our embryoid system, of the extra-embryonic (ExE) VE of the mouse embryo.

It has been recently reported that, in both the mouse embryo and in gastruloids, DE does not derive from mesendodermal progenitors that give rise to both endoderm and mesoderm but that the two lineages are already specified before ingression of mesoderm and endoderm cells at gastrulation[39]. Cell-state transitions and collective cell movements generate an endoderm-like region in gastruloids[40]. Therefore, we investigated whether we could find evidence or not of a pool of mesendodermal progenitors in our embryoids. To do so, we performed double immunostaining at gastrula stage (D4) for the endodermal marker *Sox17* and the mesodermal marker *Bra* (Supplementary Fig. 11). We did not observe any cells expressing both endodermal and mesodermal markers. This supports, in embryoids, the published observation that, in both the mouse embryo and in gastruloids, cells may already be committed at the time of gastrulation to endodermal or mesodermal fates rather than being bipotential, and representing a mesendodermal population. At D5.5, *Sox17* expression was found around multiple cavities (Fig. 3h). These cavities were surrounded by a polarized epithelium as revealed by acetylated tubulin (Fig. 3i, j) and phalloidin labelling (Fig. 3k). Expression of *Sox17* with phalloidin in this structure strongly supports the conclusion that it corresponded to the endodermal epithelium (Fig. 3k). At D6–D7, cavities aligned progressively at the midline and merged together into a single large cavity (Supplementary Movie 1) that was lined by a *Sox17* expressing endoderm-like epithelium (Fig. 3l).

Because of the lack of specific DE markers at these early stages, we were unable to independently distinguish the contribution of VE and DE to the endoderm epithelium. However, because DBA stains all VE (but not DE, mesoderm nor epiblast cells), it can be used as an indirect marker of DE formation[34]. During embryo development, intercalation of DE cells into the VE layer disperses the initially closely packed VE cells DBA(+), creating DBA unlabelled patches of cells that reveal the DE epithelium. An example of mosaic DBA(+) (VE-like cells)/DBA(−) (DE-like cells) is presented in Fig. 3m. Heterogeneity of the endoderm-like epithelium was also revealed by the patchy expression of the DE marker *Pyy* at D7 (Fig. 3n) as well as by double colour in situ hybridization for *Pyy* and *Sox17* (that labels both VE and DE) (Fig. 3o, p). In both cases, the endoderm-like epithelium in the embryoids consisted of a patchwork of VE-like and DE-like cells, similar to that of the mouse embryo.

The endodermal-like nature of the epithelium was further confirmed at D7 by looking at additional gut markers including *Cldn4*, which is expressed in the whole gut epithelium of the mouse embryo at E8.5–E9.0 (Fig. 3q), *Apela*, which is expressed in the mouse embryo in the ventral foregut on top of the heart as well as in the hindgut diverticulum (Fig. 3r) and *Rnf128*, a marker of the hindgut diverticulum (Fig. 3s). In the embryoids, the endoderm-like epithelium was clearly patterned anterior-posteriorly. At D7, this primitive gut-like tube can fold into pockets (Fig. 3t) and strongly mimicked those of the primitive gut

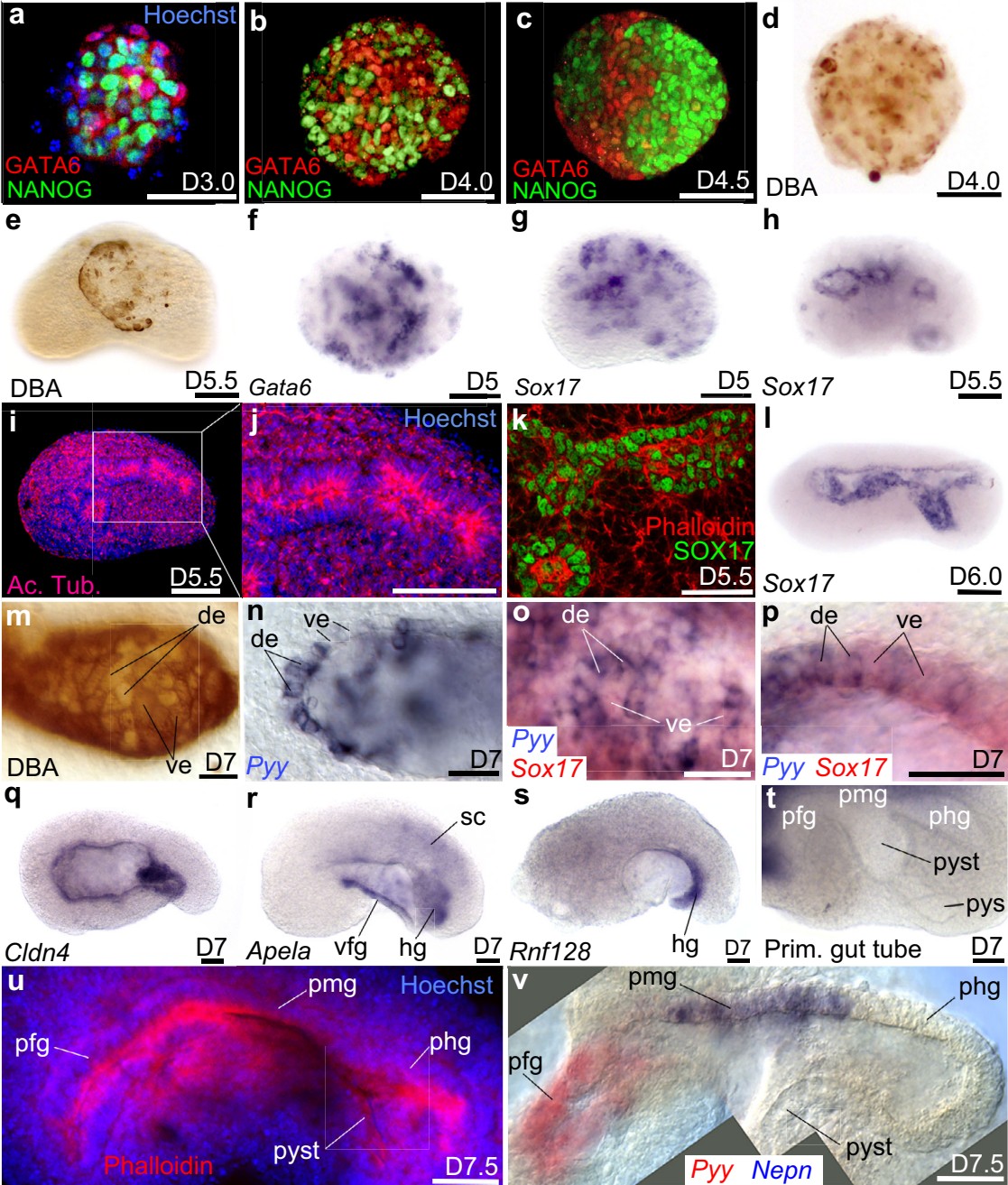

**Fig. 3 Endoderm germ layer and formation of the embryoid gut tube. a–c** Immunodetection of NANOG and GATA6 from D3.0 to D4.5. **d**, **e** DBA labelling of visceral endoderm (ve) at **d** D4.0 and **e** at the surface of the embryoid at D5.5. **f**, **g** Clumps of endoderm cells expressing **f** *Gata6* or **g** *Sox17*. At D5.5 **h** *Sox17* was expressed around small cavities bordered **i**, **j** by a polarized epithelium positive for Acetylated Tubulin (Ac. Tub). **k** Immunodetection of SOX17 identifies the endodermal-like epithelium. **l** In situ hybridization for *Sox17* at D6.0 after the merging of small cavities into a single one. **m** Mosaic of DBA(+) (ve cells) and DBA(−) cells (definitive endoderm: de) in the endoderm epithelium. **n** Patchy expression of *Pyy* (de marker). **o**, **p** Expression of *Sox17* (red) and *Pyy* (blue) showing that the endoderm epithelium was a mosaic of de and ve. **o** Superficial view of the epithelium. **p** Optical cross-section of the epithelium at higher magnification. **q–s** In situ hybridization for **q** *Cldn4* in the whole gut epithelium, **r** *Apela* in the ventral foregut (vfg) and hindgut (hg), **s** *Rnf128* in the hg. **t** Folding of the primitive-like gut showing a presumptive fg (pfg) followed dorso-posteriorly by likely presumptive midgut (pmg), presumptive hindgut (phg), presumptive yolk stalk (pyst) and presumptive yolk sac (pys). **u** Phalloidin (red) and Hoechst (nuclei, blue) labelling of primitive gut tube. **v** Combined images of a double colour in situ hybridization of pgt at D7.5 showing expression of *Pyy* (red) in pfg, *Nepn* (blue) in pmg (phg and pyst were unlabelled). sc: spinal cord. Scale bars: 100 μm except for **k**, **p**, **u**, **v**: 50 μm. Embryoids (**e–v**) in lateral views, anterior to the left, dorsal to the top. Numbers of performed experiments and of biological samples analysed is provided 'Statistics and reproducibility' in the Methods section. Variability in genes expression is provided in Supplementary Figs. 9, 10.

tube of the mouse embryo with an anterior pocket (likely foregut diverticulum) followed on the dorsal side by a straight epithelium (likely equivalent to the midgut), connecting a posterior pocket (likely equivalent to the hindgut diverticulum). This was in continuity with a ventral domain (reminiscent of the yolk stalk, and in its continuity to what may be a rudimentary equivalent of the yolk sac of the mouse embryo). This ventral domain looped at the surface of the embryoids and was connected to the anterior pocket (Fig. 3t, u). We confirmed these findings at D7.5, showing that the anterior pocket expressed the foregut marker *Pyy* (therefore identifying this pocket as the foregut-like diverticulum), followed on the dorsal side of the cavity by a domain expressing the midgut marker *Nepn* and an unlabelled pocket, likely corresponding to the hindgut (Fig. 3v) that we found to be expressing the hindgut marker *Rnf128* (Fig. 3s). On the ventral side, the epithelium often exited the embryoids forming an external loop, that may represent a rudimentary yolk sac; the territory connecting it to the primitive gut tube (Fig. 3v), may be an equivalent to the yolk stalk.

**Formation of mesodermal derivatives**. In addition to formation of a gut tube-like structure, embryoids also differentiated mesodermal derivatives. Above the primitive gut tube and below the neural plate, we identified a notochord-like structure expressing *Bra* (Fig. 4a). *Noto* (Fig. 4b) and *Foxj1* were expressed in the posterior part of this notochord-like structure. In addition, *Foxj1* was expressed in a node-like domain (Fig. 4c). Lateral to the axial mesoderm, expression of *Meox1* revealed the presence of paraxial-like mesoderm (Fig. 4d), which appeared segmented for a small fraction (11%) of *Meox1* expressing embryoids (Supplementary Fig. 12d). Segmentation of the paraxial-like mesoderm was further established by examining both the expression of *Cer1* (Fig. 4e), detected in the anterior compartment of the newly formed somite-like structures and in the adjacent anterior PSM (Fig. 4f) as well as by the expression of *Mesp1* immediately posterior to *Cer1* expression in PSM (Fig. 4f).

Lateral to the paraxial mesoderm, *Osr1* identified an intermediate mesoderm-like domain (Fig. 4g). Anterior lateral plate mesoderm derivatives were also observed, with the expression of *Smarcd3* (Fig. 4h), an early marker for cardiac progenitors as well as *Myl2* (a ventricular/cardiac muscle isoform) revealing the presence of cardiac-like tissue in the ventral anterior part of the embryoids (Fig. 4i), a location similar to that of the heart in the mouse embryo and a location and where we also observed the expression of *Mesp1* in heart-like precursors at the end of gastrulation (Fig. 2i). This cardiac-like tissue started beating at D7.5 (Supplementary Movie 2).

At D6 we found expression of *Runx1*, a transcription factor involved in primitive erythropoiesis. In the mouse embryo, *Runx1* expression is observed at E7.0–E8.0 in the proximal yolk sac mesoderm before the establishment of blood islands[41]. In the embryoids, *Runx1* expression was observed in clusters of cells loosely attached to the outer surface of the embryoids (Fig. 4j, k), likely equivalent of *Runx1* expression in the precursors of blood islands in the mouse embryo. In agreement with this hypothesis, we identified primitive erythroid-like cells expressing *Hbb-bh1*, the beta-like embryonic chain of the hemoglobulin Z (Fig. 4l, m) also located at the outer surface of embryoids within close proximity of the yolk sac.

Finally, the expression of the Kinase insert domain receptor (*Kdr*, also known as vascular endothelial growth factor receptor 2—*VEGFR-2* and as fetal liver kinase 1—*Flk1*) strongly supported the presence of a vasculature-like network in the embryoids (Fig. 4n–t). In the mouse embryo, the early mesodermal marker *Kdr* is expressed in progenitor cells of both the endocardium and

myocardium in the primitive streak[42]. These cardiogenic mesodermal cells migrate during gastrulation toward the anterior ventral part of the embryo and at the end of gastrulation *Kdr* expression is observed in endocardial tubes that arise through de novo vasculogenesis[43]. At early somite stages, *Kdr* expression is observed in the developing vasculature, which includes a pair of dorsal aortae and precursors of vitelline and cardinal veins[44]. Similar to the mouse embryo, expression of *Kdr* in the embryoids (Fig. 4n, o) was detected at D5 in a ventral anterior territory either within or near the domain of the embryoids expressing *Mesp1* (Fig. 2i) and *Smarcd3* (Fig. 4h), in tubular-like structures resembling endocardial-like tubes. The anterior position of this *Kdr* expression domain was confirmed by double colour in situ hybridization with *Bra* (Fig. 4o). At D5.5, *Kdr* expression (Fig. 4p) was observed with a pattern very similar to mouse early vascular development (cartoon in Fig. 4p), with different domains of early vascular-like development, both in the cardiac area as well as in two rods of cells aligned antero-posteriorly that may resemble the growing dorsal aortae preceded by scattered aorta progenitor cells[44,45]. Scattered cells were observed laterally, at the level of the heart. They may represent the precursors of the vitelline and cardinal veins, which appear at that location in the mouse embryo. At D6, the pattern of *Kdr* expression appeared more complex in the embryoids (Fig. 4q), showing both a high density of scattered blood vessel-like progenitors and a strong labelling of a large dorsal vessel-like structure (either dorsal aorta or posterior cardinal vein) that connected perpendicular smaller vessel-like structures, reminiscent of intersomitic blood vessels in the mouse embryo[44]. Two days later, more blood vessel-like structures were observed while less progenitors were observed and the embryoids started to be covered by a reticular network of vessel-like structures (Fig. 4r, s) that increased in density through angiogenesis. This is illustrated in Fig. 4t, which shows at high magnification, the formation of novel capillaries-like structures, likely through sprouting from pre-existing larger vessels, sprout outgrowth and sprout fusion as described for angiogenesis in the mouse embryo[46].

Altogether, these set of data demonstrate that the various mesodermal cell populations are present in the embryoids. Moreover, the position of the different mesodermal-like tissues is in good agreement with the position of their mouse embryo counterpart relative to the AP, DV and medio-lateral axes.

**Formation and patterning of the neural plate/neural tube**. The expression of *Sox1-GFP* in the anterior and dorsal parts of embryoids at D5.5 (Fig. 1) supported that embryoids also developed a neuroectoderm. Remarkably, an epithelium was morphologically visible at D6.5 on the convex side of the embryoids that was therefore defined at the dorsal side (Fig. 5a). At D7, in about 25% of embryoids made of *Sox1-GFP* ESC, this superficial epithelium expressed GFP and extended from the anterior to the posterior end of the embryoids (Fig. 5b). At this stage, we revealed the presence of *Pax6* transcripts in a single continuous domain (Fig. 5c). In the mouse embryo, two separated domains of *Pax6* are observed, in the forebrain as well as in the hindbrain and spinal cord[47]. In the embryoids, although its anterior-most expression domain appeared absent, *Pax6* expression likely extended from the hindbrain to the spinal cord (Fig. 5c). A similar observation was made in regards of to *Scube2* expression[48] (Fig. 5d). Anteriorly, transcripts of *En2*, a gene expressed in the midbrain and anterior hindbrain (rhombomere 1) of the mouse embryo[49], accumulated at the tip of D7 embryoids (Fig. 5e). We detected an anterior stripe of *Fgf8* expression, which may correspond to the isthmus, a territory at the boundary between the midbrain and hindbrain[50] (Fig. 5f). Posterior to the

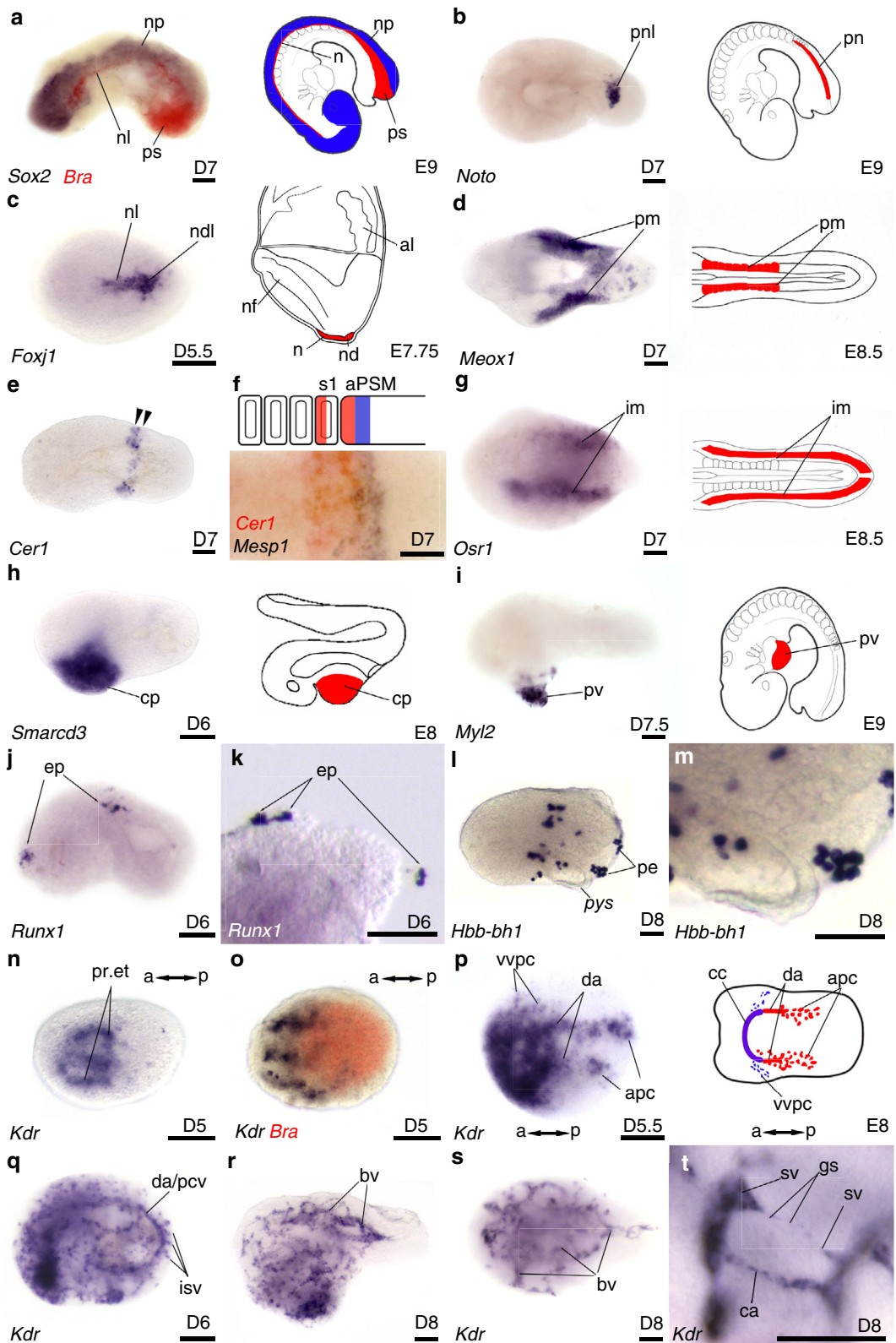

presumptive isthmus, expression of *Egr2*, a marker for hindbrain rhombomeres 3 and 5 in the mouse embryo[51], was observed within two stripes, showing a hindbrain-like identity in the neural plate of the embryoids and segmentation of the rhombencephalon into its different rhombomeres (Fig. 5g). *Fgf8* expression in an isthmus-like structure at the midbrain–hindbrain border, *Egr2* expression in the hindbrain-like structure, *Hoxd4* expression

from the posterior hindbrain-like structure, rhombomeres 6/7 to the tail[52] (Fig. 5h), *Cdx2* (expression in the posterior part of the spinal cord[53] (Fig. 5i) and finally *Hoxd9* expression in the most caudal spinal cord[54] (Fig. 5j) provides clear evidence of AP patterning in the neural plate. Finally, *Nkx1-2*, a marker for neuro-mesodermal progenitors (NMP) (located in the caudal epiblast lateral to the primitive streak of the mouse embryo[55]) (Fig. 5j)

**Fig. 4 Mesoderm germ layer derivatives in embryoids.** Single colour (**b**–**e**, **g**–**t**) or double colour (**a**, **f**, **o**) in situ hybridization revealing expression pattern for **a** *Bra* (red) in primitive streak-like (psl) and in notochord-like (nl) in contact with the neural plate (np) expressing *Sox2* (blue); **b** *Noto* in posterior part of the notochord-like (pnl); **c** *Foxj1* in nl and node-like (ndl); **d** *Meox1* likely in paraxial mesoderm (pm), **e** *Cer1* in newly formed somites (arrows). **f** High magnification of labelling with *Cer1* (red) and *Mesp1* (blue) in the anterior compartment of the newly formed somite (s1) and in the anterior presomitic mesoderm (aPSM) followed posteriorly by a stripe of *Mesp1* expression (blue). **g** *Osr1* expression likely in intermediate mesoderm (im). **h** *Smarcd3* expression likely identifying cardiac progenitors (cp). **i** *Myl2* expression in heart (pv: primitive ventricle). **j**, **k** Expression of *Runx1* in likely erythroid precursors (ep). **l**, **m** *Hbb-bh1* in primitive erythrocytes (pe) loosely attached at the surface of embryoids, pys: presumptive yolk sac. **n** Expression of *Kdr* in presumptive endocardiac tubes (pr. et). **o** Double colour in situ hybridization for *Bra* (red) and *Kdr* (blue) with anterior localization of *Kdr* expression at D5. **p** *Kdr* expression at D5.5 in various domains of early vascular-like development (summarized in the drawing on the right), da: dorsal aortae; apc: aorta progenitor cells; vvpc: presumptive precursors of the vitelline veins; cc: cardiac crescent. **q** At D6, *Kdr* expression observed in scattered vascular progenitors, in a large dorsal blood vessel (either da or posterior cardinal vein, pcv) and in perpendicular blood vessels that are likely to be intersomitic blood vessels (isv). **r**, **s** Network of blood vessels (bv) at D8 in lateral (**r**) and dorsal views (**s**). **t** High magnification of sprouting of blood vessels showing angiogenesis. ca: capillary, sv: sprouting vessel, gs: growing stem. Developmental stages of the embryoids and drawing of embryos are at the bottom right of each panel. Embryoids presented anterior to the left, in lateral view, except for **d**, **g**, **p**, **s** dorsal and **n**, **o** ventral views. Cartoons on the right in (**a**–**i**, **p**) show position of equivalent structures in the mouse embryo. Scale bars: 100 µm. Numbers of performed experiments and of biological samples analysed is provided in 'Statistics and reproducibility' in the Methods section. Variability in genes expression is provided in Supplementary Fig. 12.

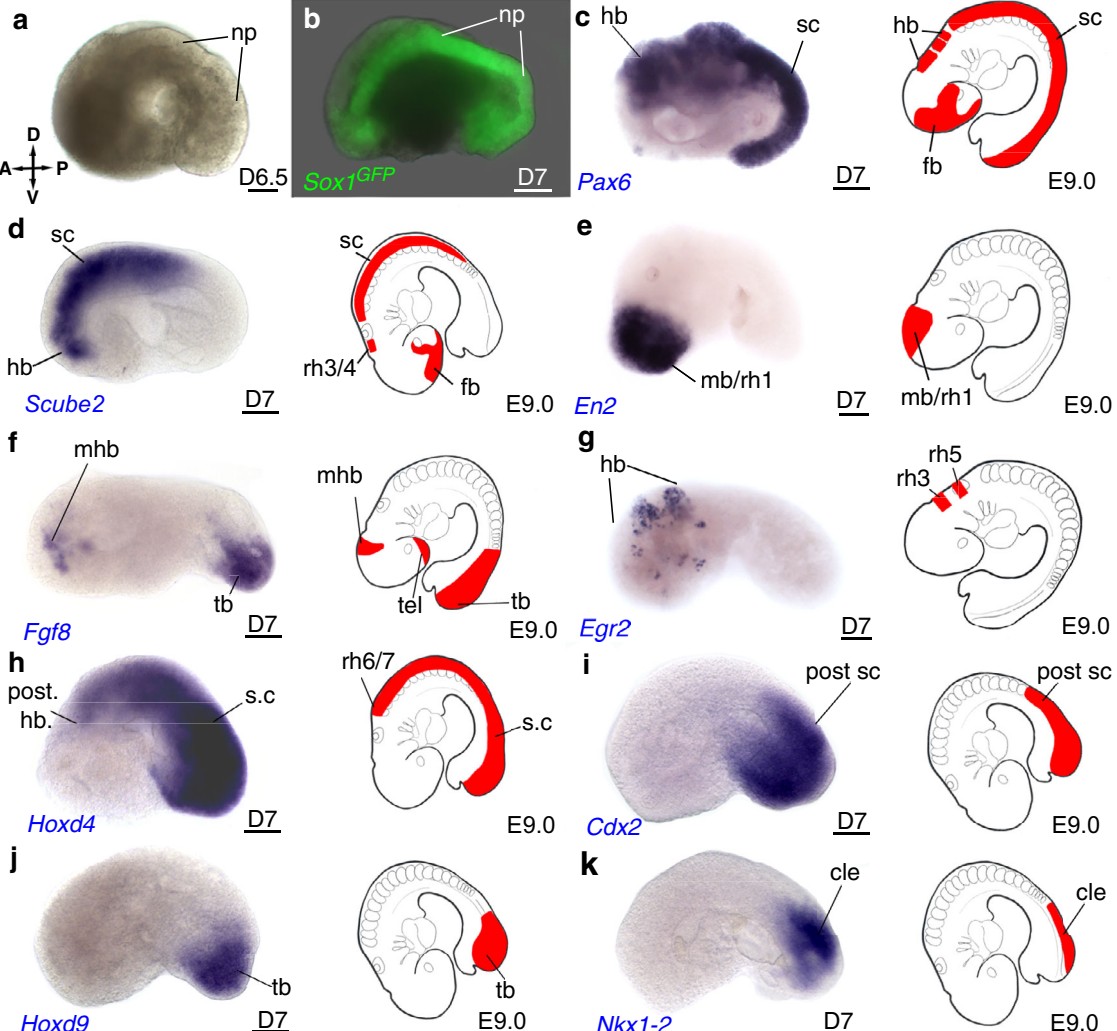

**Fig. 5 Ectoderm germ layer and antero-posterior patterning of the neural plate. a** Brightfield imaging of a representative D6.5 embryoid. np: neural plate. **b** Embryoid made of *Sox1^GFP* ESC expressing *GFP* in the neural plate. **c**–**k** In situ hybridization revealing expression of **c** *Pax6* and **d** *Scube2* in likely hindbrain (hb) and spinal cord (sc) and absence of an anterior forebrain (fb) expression domain, Rh3, Rh4 rhombomere 3 and 4. Expression of **e** *En2* in likely midbrain (mb) and rhombomere 1 (rh1), **f** of *fgf8* in an anterior stripe (likely the midbrain–hindbrain boundary, mhb) and tail bud (tb), tel: telencephalon. **g** *Egr2* in 2 stripes in hindbrain, likely in rhombomere (rh) 3 and 5. **h** *Hoxd4* in posterior hindbrain (post hb) and spinal cord, **i** *Cdx2* in posterior spinal cord (post sc), **j** *Hoxd9* in tail bud (tb), **k** *Nkx1-2* in caudal epiblast (cle). Embryoids on side view, anterior (A) to the left, dorsal (D) to the top. Cartoons on the right show expression of the corresponding gene in the mouse embryo. Scale bars: 100 µm. (V): ventral, (P): posterior. Numbers of performed experiments and of biological samples analysed is provided in 'Statistics and reproducibility' in the Methods section. Variability in genes expression and measurement of the anterior-posterior extension for the expression of *Hoxd4*, *Cdx2* and *Hoxd9*, is provided in Supplementary Fig. 13.

was found, similar to the mouse embryo, close to the posterior end of the embryoids.

About 15% of embryoids displayed a dorsal *Sox1-GFP*(+) epithelium, in these cases, we observed the folding of the neural plate-like structure around D7 (Fig. 6a). Immunostaining and in situ hybridization for *Sox2* confirmed the neuroectodermal nature of this structure (Fig. 6b, c). At D8, a frontal view of an embryoid revealed that the folded neural plate was starting to close into a neural tube-like structure for which dorso-lateral hinge points were visible (Fig. 6d). The neural plate/tube of the

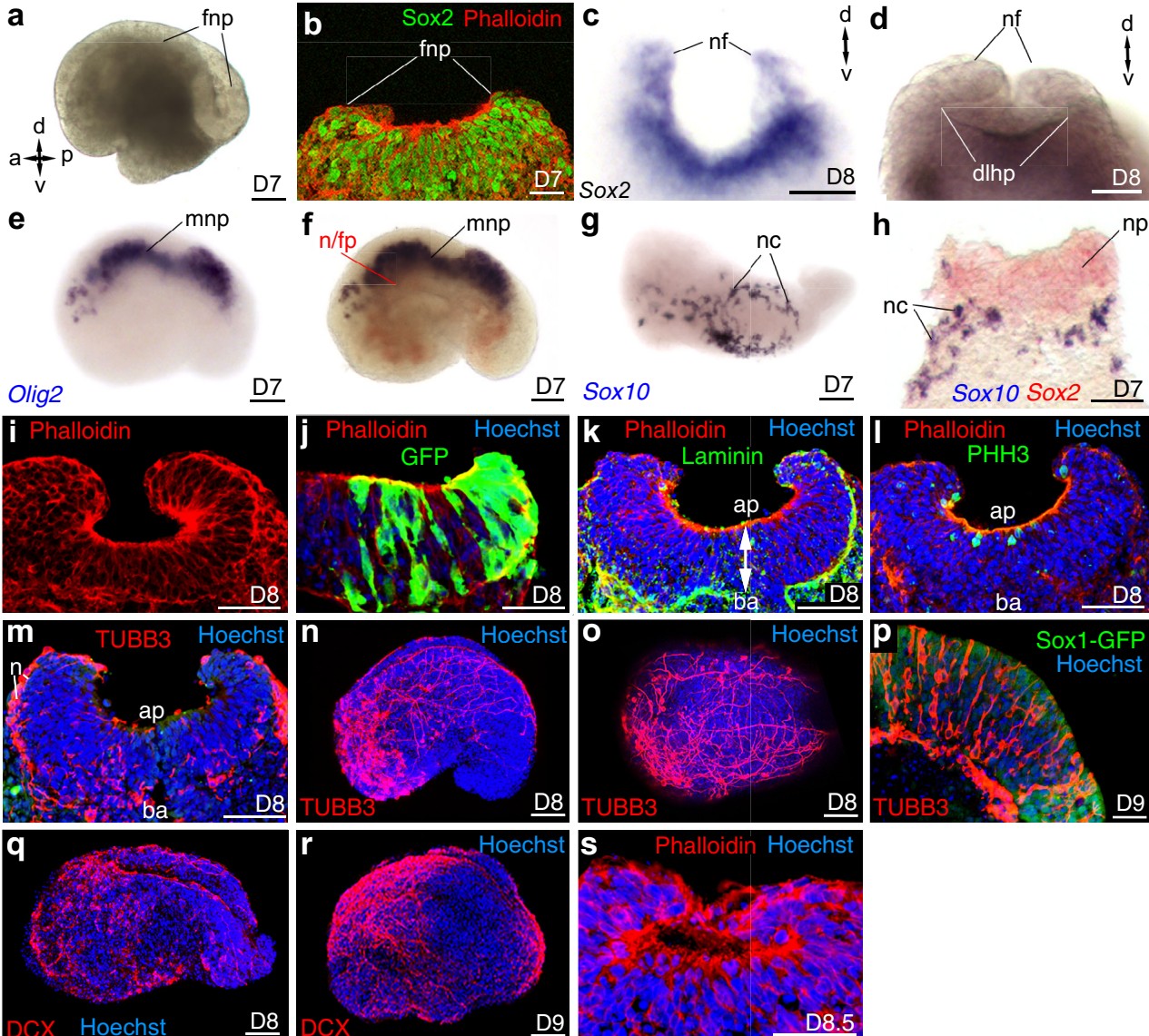

**Fig. 6 Characterization of the neural plate and formation of the neural tube. a–c** Folding of the neural plate (fnp), nf: neural fold. AP and DV axes are indicated. **b, c** Expression of *Sox2* seen by **b** immunostaining (number embryoid $N = 4/4$), **c** in situ hybridization ($N = 2/2$) on transverse section. **d** Front view of an embryoid showing the folded neural plate closing into a neural tube, dorso-lateral hinge points (dlhp). **e–h** Single or double colour in situ hybridization for **e** *Olig2* revealing motor neuron progenitor (mnp). **f** *Shh* (red) in the notochord and floor plate (n/fp) in a territory immediately ventral to *Olig2* (blue) expression domain. **g, h** Expression of *Sox10* revealing the neural crest cells (nc) in dorsal view ($N = 2$). **h** Transverse section of an embryoid after double colour in situ hybridization for *Sox2* (red—neural plate) and *Sox10* (blue—neural crest cells). **i–m** Transverse sections of a folding neural plate ($N = 9/9$). **j** The organization of a pseudostratified columnar epithelium. Green: *GFP* cells extending from the apical to the basal side of the epithelium ($N = 3/3$) **k** apical (ap)–basal (ba) polarity of the epithelium with apical accumulation of actin labelled with Phalloidin and basal accumulation of laminin ($N = 3/3$). **l** Immunolabelling for Phospho-Histone H3 (PHH3) revealing that cell divisions occur at the apical side of the epithelium (number of embryoids $N = 3/3$, number of mitotic cells at the apical side $N = 7/8$). **m** Immunolabelling revealing the presence of neuronal cell bodies (n) expressing TUBB3 at the basal side of the neural plate ($N = 3/3$). **n, o** Expression of TUBB3 (red) showing neurons extending their axons in the neural plate and toward the non-neural part of the embryoids (**n**, lateral view; **o**, dorsal view of the same embryoid). **p** TUBB3 expressing neurons (red) in multiple layers of the anterior neural plate in a parasagittal section at D9 (*Sox1-GFP*, green) ($N = 2/2$). **q, r** Expression of DCX (red) confirming the presence of neurons and axons extending both within the neural plate and toward the non-neural part of the embryoids ($N = 4/4$). **s** Neural plate closed into a neural tube ($N = 2/8$). **g–s** Hoechst is labelling the nuclei. Developmental stages of the embryoids and drawing of embryos bottom right of each panel. Scale bars: 100 μm. Numbers of performed experiments of biological samples analysed and variability in genes expression are provided in Supplementary Fig. 14.

embryoids was also patterned along its Medio-Lateral/DV axis. Ventrally, we found transcripts of *Olig2*, a marker for progenitors of motor-neurons and oligodendrocytes (Fig. 6e, f), a gene whose expression is induced by SHH morphogen that is secreted by the notochord and floor plate[56] and whose transcripts were observed in the embryoids, located ventrally to the *Olig2* expression domain and above the endoderm-like epithelium (Fig. 6f). Dorsally, expression of the neural crest cell marker *Sox10*[57] (Fig. 6g, h) occurred along the lateral borders of the neural plate-like structures, as is the case for authentic mouse embryo.

We further analysed the neurulation process by preparing transverse sections of the embryoids. At D8, sections confirmed that the neural plate-like structure had folded (Fig. 6i). Following the strategy used in the mouse embryo[58] and using mosaic embryoids made of two strains of ESC (10% of 129 ESC that express GFP constitutively and 90% of unlabelled E14TG2 ESC) we showed that the neural plate-like structure was a pseudostratified columnar epithelium with individual neuroectodermal cells extending from its apical to basal surface (Fig. 6j). This neuroepithelium exhibited proper apical–basal polarity with apical actin filaments and extracellular matrix containing laminin at its basal side (Fig. 6k). Similar to a typical vertebrate neuroepithelium, in the embryoids, mitosis occurred at or near the apical surface (Fig. 6l) while cell bodies of neurons expressing class III β-tubulin (TUBB3) were located basally (Fig. 6m). At D8, TUBB3 expressing neurons extended their axons both within the developing neural plate/tube and toward the non-neural part of the growing embryoids (Fig. 6n, o) and at D9, in the anterior part of the neural plate, TUBB3 expressing neurons appeared in multiple cell layers (Fig. 6p), which were reminiscent of those observed at the onset of cortical neurogenesis in the hindbrain of the mouse embryo[59]. As well, at D8-D9, *Doublecortin* (DCX), a marker for early post-mitotic neurons, was expressed in neurons extending their axons both along the neural tube and toward the non-neural part of the embryoids (Fig. 6q, r). Finally, at D8.5, the neural plate closed into a neural tube (Fig. 6s).

In summary, at a stage equivalent to the neurula of the mouse embryo, the embryoids display a neural plate/tube-like structure, with AP and DV axes, showing the histological characteristics of a typical mammalian neural plate/neural tube.

**Origin of cells giving rise to the different germ layers**. To identify whether cells of the different germ layer derivatives originated from the signalling centre or from the naive aggregate, we generated embryoids with combinations of *GFP* labelled ESC (129-GFP that express *GFP* constitutively) and unlabelled E14TG2a ESC (Supplementary Fig. 15). First, we labelled the signalling centre using cells constitutively expressing the GFP. The resulting embryoids showed an unlabelled neural plate while *GFP* expressing cells were for the vast majority found in both the mesodermal and endodermal layers (and possibly in the neuro-mesodermal progenitor population in the posterior part of the embryoids). Focusing on the endoderm we found that the entire endoderm epithelium was fluorescent. Double immuno-fluorescence (Supplementary Fig. 15d) for GFP and SOX17 revealed the presence of SOX17(+) cells from the naive (unlabelled) part of the embryoid that were likely VE cells deriving from GATA6(+) cells present in the naive part of the embryoid (Fig. 3b, c) as well as SOX17(+)/GFP(+) cells from the signalling centre that corresponds to both VE cells from that domain and DE induced there by the high level of NODAL signalling. Conversely (Supplementary Fig. 16), when the naive territory was *GFP* labelled, we observed an accumulation of fluorescence in the ectoderm, in particular in the neural plate. In addition, by focusing on the endoderm epithelium, we confirmed the

contribution of cells from the naive territory to the formation of the gut tube. These results demonstrate that cells from the primitive streak-like structure of the embryoids gave rise to mesoderm and endoderm, while naive cells gave rise to ectoderm (as well as a small population of mesodermal cells induced in the naive territory by the morphogens secreted by the signalling centre, Fig. 1i, Fig. 2e).

In summary, anterior neuroectodermal cells were derived from the naive territory, VE was derived from both the signalling centre and the naive territory, and definitive endoderm and most of the mesoderm was derived from the signalling centre. The origin of the neuromesodermal progenitor population has not yet been established. This cell population may well be within the population of *Bra* expressing cells present in the naive territory (Figs. 1i and 2e).

**Transcriptome analysis of developing embryoids**. To complement our analysis of cell types, tissues and organ primordia present in the developing embryoids, we performed transcriptomic analyses at the neurula-stage (both bulk and single-cell transcriptomics). Bulk transcriptomes of D7 and D8 embryoids were compared to published transcriptomes of mouse embryos at stages in between E8.0 and E9.5[60] (Fig. 7). Principal component analysis (PCA) was performed on transcriptome data from all replicates for each of the following developmental time points embryos at E8, E8.5, E9, E9.5 and embryoids at D7 or D8 (Fig. 7a). These analyses revealed noticeable differences in the transcriptomes, which was expected because we had already determined that the embryoids lacked forebrain and anterior midbrain structures (Figs. 1–6). PCA revealed that the transcriptomes of D7 and D8 embryoids were more similar to the transcriptomes of mouse embryos at stages E8.5–E9.0 (Fig. 7a). This was in agreement with the data we had assembled using in situ hybridization and immunostaining of various tissues and germ layer derivatives (Figs. 1–6). This analysis confirmed that embryoids expressed genes specific for tissues and cell derived from all three germ layers (Fig. 7b–d), although the level of expression was slightly lower in embryoids compared to neurula-stage mouse embryos. The most remarkable difference between mouse embryos and embryoids was observed for anterior-most neuroectodermal genes that were either not expressed (e.g. *Six3* and *Six6*, two forebrain specific genes) or expressed in very low amounts (e.g. *Dmbx1*, a midbrain specific gene) in the embryoids. This was in agreement with our in situ/immunolabelling analyses showing that the embryoids lacked anterior structures. We also observed that while present, RNAs coding for markers of lateral plate mesoderm (*Foxf1*) and intermediate mesenchyme (*Osr1*) were lower in embryoids than in mouse embryos. This suggested that the embryoids may develop less lateral mesoderm than the embryos.

Bulk transcriptomes of D7 and D8 embryoids were also compared to transcriptomes of D7 and D8 controls (made by merging two untreated ESC aggregates at D3). PCA of embryoids and control transcriptomes revealed that the experimental replicates were highly reproducible and strongly clustered (Supplementary Fig. 17a). Each condition and stage segregated in distinct groups indicating that embryoids and controls had distinct transcriptomes.

To identify the biological functions of genes upregulated in embryoids or in controls, we determined functional categories of each gene by querying the Gene Ontology database[61,62]. Functional grouping of the GO terms based on GO hierarchy revealed that amongst the most prominent groups of genes upregulated in embryoids at D8 were those associated with the development of mesoderm (vasculature, heart, muscle) and

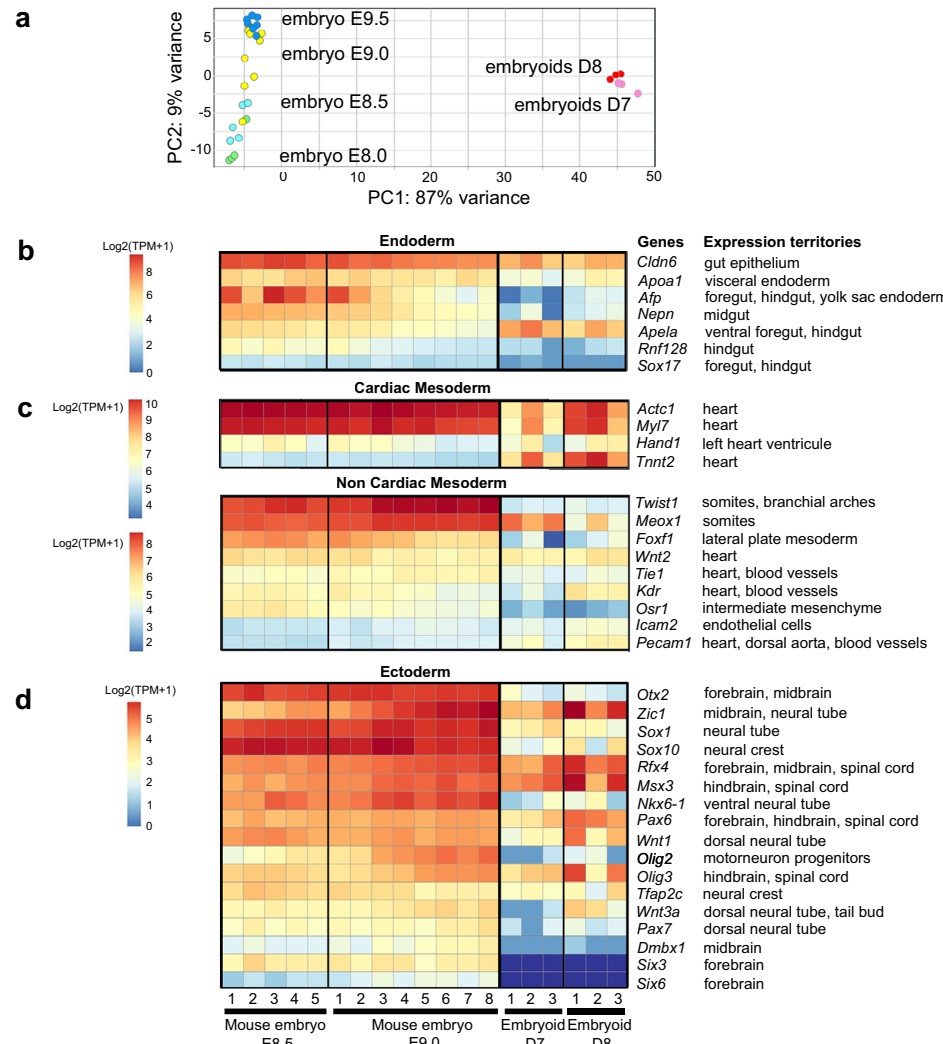

**Fig. 7 Comparison of transcriptomes of D7 and D8 embryoids with of E8.5 and E9.0 mouse embryos. a** Principal component analysis (PCA) of bulk RNA-seq data from 3 replicates of 10 embryoids at D7 (pink) and D8 (red), 4 replicates of mouse embryos at E8.0, 5 replicates at E8.5, 8 replicates at E9.0 and 7 replicates at E9.5 showing that the embryoids bulk-RNA transcriptome is similar to the one for mouse embryos at E8.5 and E9.0. **b–d** Heatmaps of normalized expression in D7 and D8 embryoids and in E8.5 and E9.0 mouse embryos for gene markers of germ layer derivatives in the mouse embryo. **b** Endoderm, **c** Mesoderm, **d** Ectoderm. Because of their different levels of expressions, comparison of the expression of cardiac genes between mouse embryos and embryoids has been performed independently of the analysis for non-cardiac mesoderm. The replicates represented in these graphs were derived from biologically independent samples. For each gene, the expression territory (right) corresponds to the main expression domain of the gene in the mouse embryo between the embryonic stages TS13-TS15 (from 8.0 to 10.25 dpc) described in the Mouse Genome Informatics (MGI) Gene Expression Database (GXD) at informatics.jax.org/expression.shtml.

endoderm (digestive system, respiratory system) derivatives as well as those associated with tube development and tube closure (Supplementary Fig. 17b-d). Conversely, groups of genes upregulated in controls were associated with the formation and patterning of anterior brain (forebrain and midbrain) but not with endoderm, mesoderm or other more posterior neural tissue (Supplementary Fig. 17b-c, e).

Similar observations were made by examining the expression level of markers for the three germ layers. We found expression of endoderm (Supplementary Fig. 18a) and mesoderm (Supplementary Fig. 18b) specific genes in embryoids but not in controls. In the neuroectoderm, genes expressed in hindbrain and spinal cord were observed only in the transcriptomes of embryoids while transcripts for genes specific for anterior CNS domains (forebrain and midbrain) were found in controls (Supplementary Fig. 18c). In summary, cells from control aggregates displayed anterior neuroectodermal identity, in accordance with the neural

induction model in which forebrain fates constitute a default positional identity in the early neuroectoderm[63,64]. Cells of D7 and D8 embryoids did not show any anterior-most neural identity but we observed a wide variety of genes normally found present in the mouse embryo, from the isthmus/midbrain–hindbrain boundary to the tip of the spinal cord.

To further characterize the identity of cells and tissues that are present in growing embryoids, we performed single-cell RNA sequencing (scRNA-seq). To do so, 10,973 cells from 90-pooled D8 embryoids were isolated and their RNA sequenced. The resulting dataset was then compared to previously published scRNA-seq datasets of mouse embryos (E7.5, E7.75, E8.0, E8.25, E8.5[65] and E9.5, E10.5[66]). To identify which of these 7 developmental stages had a single-cell transcriptome closest to that obtained from the transcriptome of D8 embryoids, 5000 cells were randomly selected from each scRNA-seq dataset (D8 embryoids and embryos at the 7 developmental stages) and

aggregated into a "bulk like RNA-seq" dataset. The expression count of these 8 datasets was normalized and sample-to-sample distances were calculated and visualized by PCA (Fig. 8a). Normalized expression counts of the different datasets were log2 transformed and used to calculate a Pearson correlation coefficient. A correlation plot was utilized to compare the embryoids dataset with that obtained from mouse embryos at stages E7.5–E10.5 (Fig. 8b). In this plot, the highest correlation coefficient between mouse and embryoid datasets with that obtained was for E8.5 mouse embryos, which was therefore selected for our comparative analysis of embryos and embryoids transcriptome data.

To compare cell populations present in E8.5 mouse embryos and D8 embryoids, both transcriptomes were integrated and then analysed using unsupervised clustering and visualized using UMAP. Cluster identity was inferred using differentially expressed genes (DEG) (Supplementary Data 1) and molecular markers published in previous analyses of mouse embryo transcriptomes (Supplementary Table 2). This allowed us to identify 24 cell clusters in the single-cell transcriptome of the mouse embryo (Fig. 8c) and we found 23 of them in the embryoids; the surface ectoderm cluster was absent from the embryoid transcriptome, Supplementary Table 3).

On UMAP, the larger clusters comprised clusters for neuromesodermal progenitors, spinal cord, midbrain–hindbrain that were organized into a continuum of states recapitulating spatiotemporal features of the developing mouse embryo at E8.5. This was also true for the mesoderm in which the larger clusters comprised clusters for NMP, presomitic mesoderm (PSM), paraxial mesoderm (PM), somite, pharyngeal mesoderm and heart. Additional large clusters comprised neurons, gut, notochord, floor plate and haematoendothelial progenitors. Smaller clusters included intermediate mesoderm, definitive endoderm, forebrain, extra-embryonic mesoderm, neural crest, amnion and endothelium. Finally, because of their very low cell numbers (Supplementary Table 3) allantois, extra-embryonic endoderm and erythrocytes could not be considered as true cell clusters in the sc-trancriptome of embryoids. However, the cells present were highly similar in their transcriptome to cells of equivalent clusters in the mouse embryo.

Cluster identification was supported by analysis of specific markers expression visualized on UMAP (Fig. 8c) or in dot plot graph (Fig. 8d). For the majority of clusters, we observed a strong similarity between embryoids and embryos for their position on UMAP (Supplementary Fig. 19), in the percentage of cells expressing the marker analysed and in the intensity of its expression (Fig. 8c). Extended dot plot data for additional markers defining the 24 clusters presented in (Supplementary Figs. 20, 21) allowed us to reach the same conclusions.

We also performed developmental differentiation trajectory analysis by pseudotime (Fig. 8e). In the mouse embryo at E8.5, NMP of both spinal cord and trunk/tail mesoderm are present in the tail bud and are the progenitors for both neural and mesodermal cells for these posterior territories[67]. Along these two differentiation trajectories, cells are ordered from the less differentiated cells in the NMP domain to the most differentiated cells and this is highly correlated with the position of cells along the posterior to anterior axis. Similarly, in the embryoids we found (Fig. 9) the progenitors for the posterior and truncal mesoderm expressing *Nkx1*-2 then expression of genes associated with intermediate level of differentiation such as *Mesp1*, *Cer1*, *Meox2* and *Pax9*, and then terminal/advanced differentiation markers such as *Nkx2-5* and *Etv2*. The succession of gene expressions along the pseudotime axis reflected the differentiation state of mesodermal cells with the anterior-most territories (e.g. heart) differentiating first while more posterior territories

(e.g. paraxial mesoderm) differentiate later and the posteriormost territory, the NMP cells, that generates the mesoderm of the posterior end of the embryoids, are only at an early differentiation state and differentiate last. Both the correlation between pseudotime differentiation of each cluster/subcluster analysed and the posterior to anterior position of the expression domains of specific markers (identified previously by in situ hybridization —Figs. 1, 4, 5) was consistent all along the AP axis from the NMP cluster posteriorly to the haematoendothelial progenitor and endothelium clusters anteriorly. The same observations were made for the neural pseudotime differentiation trajectory (Fig. 10a) with the anterior-most territories (forebrain, midbrain–hindbrain) differentiating first while more posterior neural territories including spinal cord as well as the NMP, that give rise to the posterior neural tissues, differentiate later.

The analysis of the expression territory of neuroectodermal markers (Fig. 5) and clusters of cells expressing them (*Pax6* in spinal cord and midbrain–hindbrain cluster, *Cyp26b1* in the midbrain–hindbrain cluster, *Otx2* in the Forebrain cluster, Fig. 10c–e) from the spinal cord to the forebrain, confirmed that embryoids displayed a patterned AP axis. To test whether the neural territory established cell states segregated along the DV axis, we combined floor plate, spinal cord and midbrain–hindbrain into a single neural cluster (FP-SC-M-HB, Fig. 10a) and analysed for the presence of ventral, intermediate and dorsal cell states within this neural cluster (Fig. 10f–v). Remarkably, we found cell states specific for the dorsal part of the mouse embryo neural tube (*Wnt1*, *Msx3*, *Zic1*, *Pax7*, *Pax3*) clustered on one side of the FP-SC-M-HB cluster while cells expressing ventral specific states (*Nkx2-9*, *Olig2*, *Foxa2*, *Shh*, *Nkx6-1*) were found clustering on the opposing side. Cells displaying intermediate states in the neural tube (*Dbx2* and *Pax6* that is expressed dorsally but excluded from the most ventral position) were found clustered in between domains with ventral and dorsal cell states. We also found that the neural crest cluster expressing *Sox10* was close to the domain of the FP-SC-M-HB group with dorsal cell states. Cells of the floor plate cluster expressing *Shh*, were identified close to cells with the ventral state. Altogether, this analysis revealed that the neural tube-like structure seen in the embryoids was also patterned along its DV axis in a manner similar to the DV patterning of the mouse embryo.

In summary, the bulk and single-cell transcriptomic analyses confirmed and extended our observation that embryoids undergo extensive patterning in the three germ layers along AP and DV axes. Moreover, our scRNA-seq analyses revealed that our embryoids execute developmental programmes in a spatiotemporal manner similar to that of the mouse embryo.

## Discussion

Most strategies used so far for building embryonic-like structures from mammalian ESC rely on the self-organization properties of different cell lines in various culture media. For example, mixing ESC of different origins (embryonic and extra-embryonic) allowed the formation of structures similar in shape with gastrulating mouse embryos but these failed to develop to neurulation stage. Relying solely on the ability of these cells, when placed in contact with each other to exchange signals and to turn on and regulate developmental programmes gives limited opportunities for the experimental control of development.

The second approach used relies on activation of the WNT/β-catenin signalling pathway by a pulse of WNT agonist. Surprisingly, this pulse of WNT agonist added to the culture medium, while affecting the stem cell aggregate uniformly results in the formation of an asymmetric centre expressing mesoderm

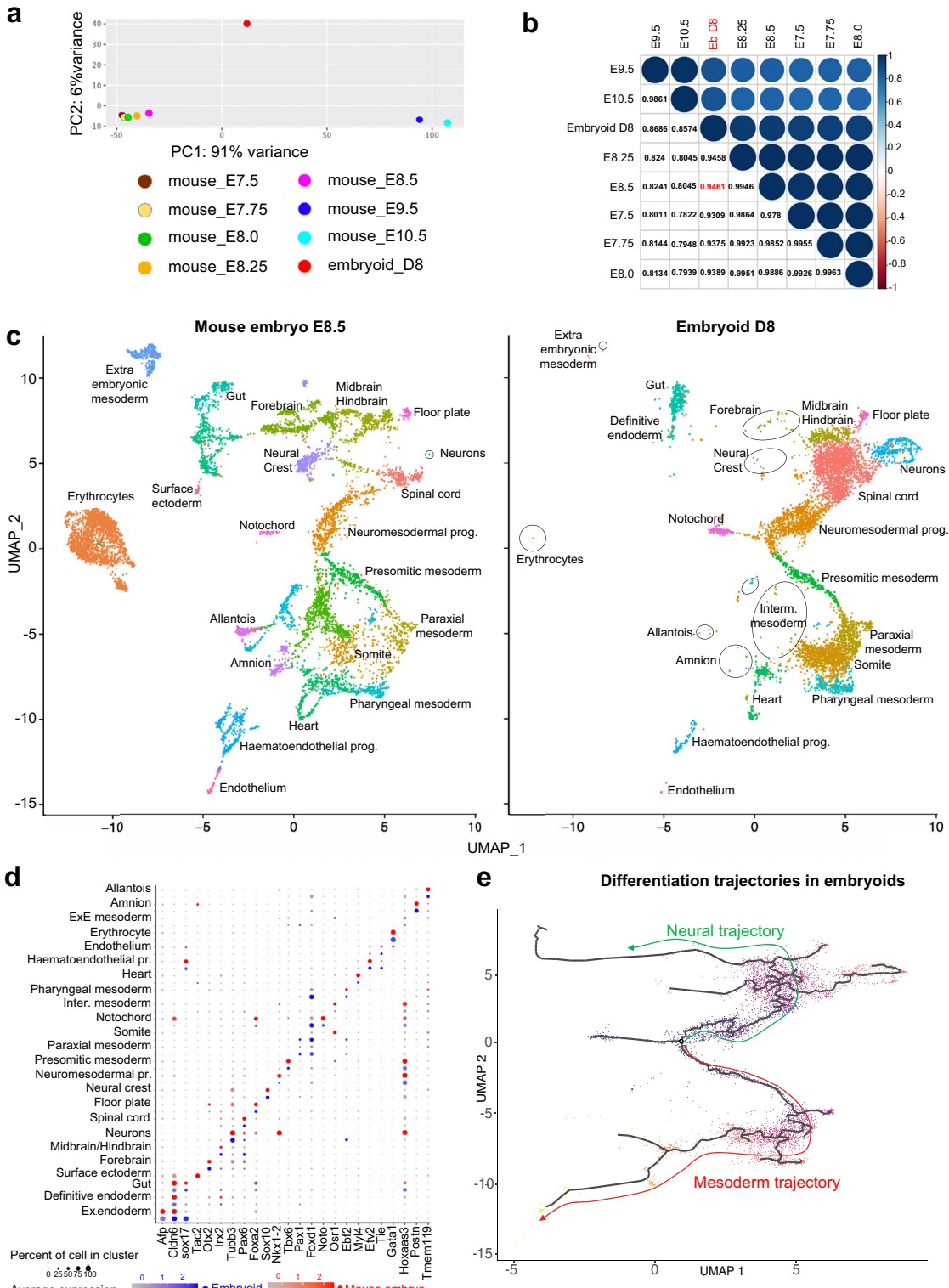

**Fig. 8 Comparison of embryoids and of mouse embryos scRNA-seq transcriptomes. a** PCA of bulk RNA-seq generated from 5000 randomly selected cells from scRNA-seq of embryoids at D8 and mouse embryos at E7.5, E7.75, E8.0, E8.25, E8.5, E9.5 and E10.5. **b** Pearson correlation coefficient plot for D8 embryoids and mouse embryos in between E7.5 and E10.5 showed the highest correlation between the transcriptome of D8 embryoids (ebD8) and of E8.5 mouse embryos. **c** UMAP plots showing 6639 cells isolated from D8 embryoids and 8590 cells isolated from E8.5 mouse embryos. A unique colour was used to show cells of each cluster (prog.: progenitors, Inter.: intermediate). The surface ectoderm cluster was only present in mouse embryos. Clusters with a small number of cells were delineated by circles or by an oval line. **d** Dot plot graph showing both the percentage of cells (diameter of the dot) of a given cluster that expressed the marker gene of that cluster and its level of expression (gradient of colour, blue for embryoids, red for mouse embryos). Names of the 24 clusters are given at the left and name of the marker gene for each cluster are given at the bottom. For each cluster, mouse embryo data are given in red top and the embryoid data are given in blue. **e** Pseudotime differentiation trajectory analysis for neural and mesoderm trajectories in D8 embryoids.

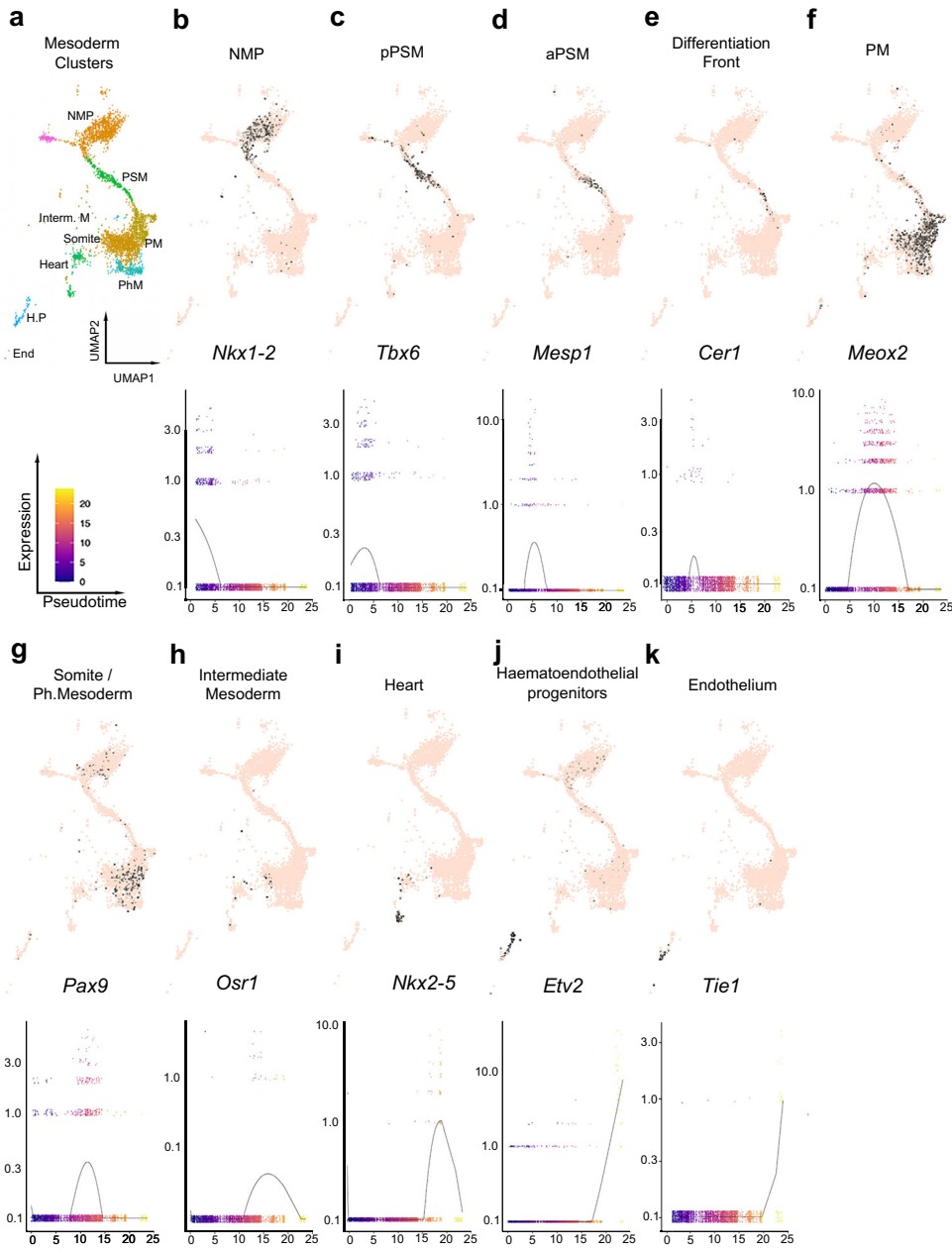

**Fig. 9 Mesoderm clusters, subclusters and mesodermal pseudotime differentiation of their specific marker genes. a** Selected mesoderm clusters: NMP: neuromesodermal progenitors, PSM: presomitic mesoderm, PM: paraxial mesoderm, Interm. Mesoderm: intermediate mesoderm, somites, PhM: pharyngeal mesoderm, heart, hematopoietic prog. (progenitors), endothelium. **b–k** UMAP plot (top) and spline plot (bottom) representing changes of expression over pseudotime for marker genes of each cluster and subcluster: **b** *Nkx1-2* for NMP, **c** *Tbx6* for posterior presomitic mesoderm (pPSM), **d** *Mesp1* for anterior PSM (aPSM), **e** *Cer1* for the differentiation front, **f** *Meox2* for PM, **g** *Pax9* for somite and pharyngeal mesoderm, **h** *Osr1* for intermediate mesoderm, **i** *Nkx2-5* for heart, **j** *Etv2* for haematoendothelial progenitors, **k** *Tie1* for endothelium.

identity. This spontaneous breaking of symmetry with a self-organization of a posterior signalling centre likely depends on the ability of some cells in which the WNT/β-catenin signalling has been activated to organize themselves into a cluster expressing WNT3 and NODAL. One possible mechanism could be that stimulation by CHIR99021 activates the WNT pathway at a higher level in cells at the surface of the aggregate (and therefore in direct contact with the culture medium containing the agonist) than in the deep cells of the aggregate. A high enough level of stimulation may result in the expression of NODAL and induction of mesoderm in superficial cells while deeper cells, receiving

less stimulation by CHIR99021, may retain ectoderm identity. It is known that mesodermal and ectodermal cells differ in cell adhesion and cortical tension, properties that are described in cell sorting to allow controlling cell–cell contact formation: adhesion increasing the contact size and cortex tension decreasing it[68]. Due to these properties, cell sorting may occur in the aggregates, resulting in the formation of two distinct groups of ectodermal and mesodermal cells, with the latter cells expressing the posteriorizing morphogens WNT3 and NODAL. However, the initial pulse of WNT agonist not only induces mesoderm but it stimulates, to some degree, the deep cells of the aggregates. This

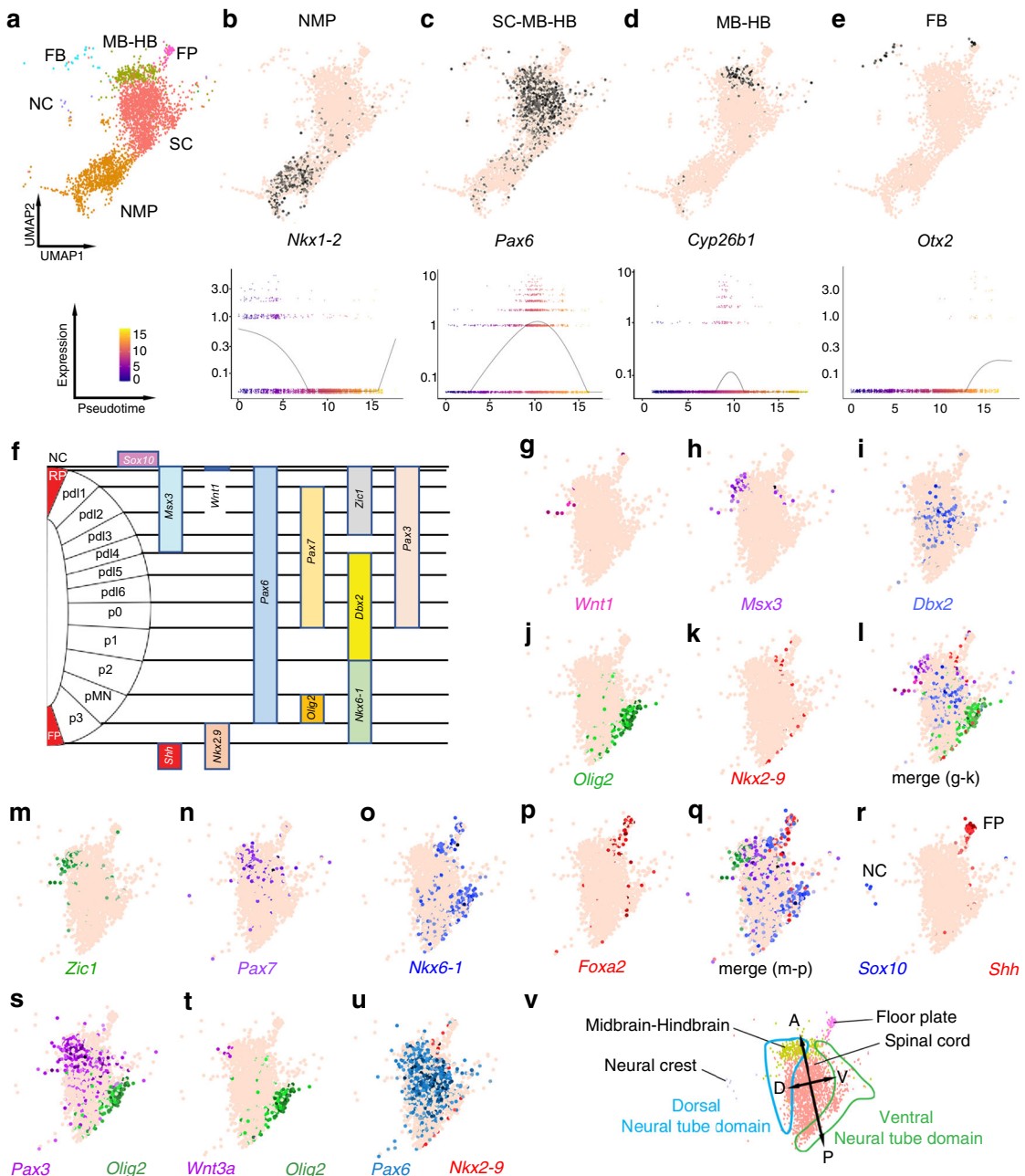

**Fig. 10 Neurectoderm clusters and neural pseudotime differentiation of their specific marker genes and segregation of cell states along the dorsoventral axis of the neural tube. a** Selected ectoderm clusters: FB: forebrain, MB-HB: midbrain–hindbrain, FP: floor plate, SC: spinal cord, NMP: neuromesodermal progenitors, NC: neural crest. **b–e** (top) UMAP plot and (bottom) spline plot representing changes of expression over pseudotime for marker genes of each cluster: **b** *Nkx1-2* for NMP, **c** *Pax6* for SC, MB-HB, **d** *Cyp26b1* for HB, **e** *Otx2* for FB. **f** Schematic of the expression pattern of genes expressed in domains of the mouse embryo neural tube along its DV axis. Left: one half of the neural tube in cross-section. From dorsal to ventral: NC: neural crest (outside of the neural tube), RP: roof plate, then arrayed along the DV axis of the neural tube are various progenitor domains: pdl1–pdl6, p0–p2, pMN (motor neuron), p3 then ventrally FP: floor plate. Domains of expression of the markers used to analyse the segregation of cells of neural clusters along AP axis are indicated as bars on the right. **g–k** Coloured cells expressing specific markers of the dorsal neural tube domain: *Wnt1*, *Msx3*; intermediate domain: *Dbx2*; ventral domain: *Olig2*, *Nkx2-9*. **l** Merge of all cells expressing these five marker genes. **m–p** Coloured cells expressing specific markers of dorsal domain: *Zic1*, *Pax7*, ventral domain: *Nkx6-1*, *Foxa2*, **q** merge of coloured cells expressing these four marker genes. **r** Position of NC and FP clusters expressing, respectively, *Sox10* and *Shh*. **s–u** Merge of coloured cells for one dorsal and one ventral markers: **s** *Pax3* (dorsal) and *Olig2* (ventral), **t** *Wnt3a* (dorsal) and *Olig2* (ventral), **u** *Pax6* (dorsal) and *Nkx2-9* (ventral). **v** Summary of clustering of cells expressing dorsal and ventral specific markers, clearly showing the segregation of cell states along the DV axis of the neural tube in a manner similar to what is observed in the mouse embryo.

may explain why the development of gastruloids is limited to post-occipital structures, likely due to the posteriorizing activity of WNT.

Our study describes an alternative strategy to generate embryonic-like structures in a more experimentally controlled manner and one that does not rely on self-organization of the aggregates. In order to break the spherical symmetry of ESC aggregates, we engineer a local signalling centre secreting WNT and NODAL morphogens and we use the property of ESC aggregates to spontaneously merge together when placed into

contact, with the goal of creating an asymmetric morphogen secreting centre into the resulting structure. This approach has a number of advantages over the formation of spontaneous signalling centres because we can control: (a) the time at which the signalling centre is used to instruct the naive aggregate, (b) the size of the signalling centre and (c) the intensity of stimulation (BMP4 concentration, duration of instruction). While this was not the original purpose of the present study, we illustrate that by modifying one parameter in our instruction protocol (e.g. three different concentrations of BMP4) (Supplementary Fig. 22), such a modification impacts erythrocytes, heart and gut formation, in terms of both their frequency in the embryoid population and/or organ size. (d) Using this method, we can also generate mosaic of embryoids in which cells from the naive aggregate and from the signalling centre come from different lines.

This strategy for instructing ESC with an asymmetrically located morphogen allows for the initiation of developmental programmes that through downstream response factors leads to the successive developmental steps that are to those occurring during normal mouse embryonic development. We have documented the formation of all three germ layers through a gastrulation process. We describe movements of cells that are exposed to a high level of morphogen activity (cells from the signalling centre that move from posterior to anterior), as well as movements more distant cells originating from the naive part of the embryoid, that like in zebrafish or *Xenopus* first move vegetally/posteriorly before ingressing or involuting, reversing their direction and moving toward the anterior/animal pole of the embryo. Using multiple approaches (in situ/immunolabelling, bulk and scRNAseq) we analysed the formation and differentiation of the three germ layer derivatives, that display clear AP and DV patterning. Remarkably, we document events during the early stages of embryoid development that are very similar to the formation of the primitive endodermal cell population of the mouse embryo at the blastula stage. Later in development, we find evidence of an endoderm-like epithelium, made of a mosaic of visceral and definitive endoderm and that expresses markers specific for different regions of a primitive gut-like tube. This embryoid primitive gut-like tube folds into foregut and hindgut diverticulae and in a fraction of them, display various regions of the gut tube epithelium, including structures similar to yolk stalk and yolk sac.

Embryoids also differentiate mesodermal derivatives. As the phylum Chordata is defined by the presence of an axial mesoderm structure, the notochord, a true in vitro model for a vertebrate embryo should include notochord. Indeed, we observe a notochord-like structure expressing *Bra*, *Chrd*, *Foxj1* and *Shh* in our embryoid model. The notochord extends from posterior to anterior in the axis of all vertebrate embryos, underneath the neural plate and above the endoderm epithelium. This is the location we found in our embryoid model for this structure. Lateral to this axial mesoderm, we also identify segmented paraxial mesoderm and intermediate mesoderm-like domains, anterior lateral plate mesoderm-like derivatives including a cardiac tissue/heart, that beats by D7.5, primitive erythrocytes cells and blood vessels developing through both the processes of vasculogenesis and angiogenesis. Finally, we characterize an ectodermal layer that differentiates into a neural plate-like structure, displaying histological properties found in the neuroepithelium of the mouse embryo (e.g. pseudostratified columnar epithelium, apical–basal polarity, mitosis occurring at or near the apical surface, neurons). When properly placed on the dorsal side, as seen in a subset of the embryoids we also observe its folding and closure into a neural tube. Embryoids develop from the posterior midbrain to the tip of the caudal

region and the neural plate exhibits AP and DV patterning. These data acquired using immunolabelling and in situ hybridization were confirmed by our analysis of the scRNA-seq data. Altogether, our embryoid model shows substantial advances in an effort to reproduce mammalian embryonic development in vitro, mainly by its capacity to develop the three germ layers properly organized along the AP axis, allowing for the different domains receiving and/or sending signals to and from the surrounding tissues.

However, the embryoids described in this study are not yet a complete model of mammalian embryos. The CNS of the embryoids lacks the anterior-most brain domains, the forebrain and part of the midbrain. Their absence can be explained by the two-step model of neural induction, known as the activation–transformation model[62,63]. According to this model, the neural tissue, when initially induced, acquires a forebrain identity. Forebrain fates constitute a default positional identity in the early neuroectoderm. In the transformation step, part of the neuroectoderm is converted to posterior fates by the activity of signals that specify midbrain, hindbrain and spinal cord. With our current strategy to induce embryoid development we probably do not provide any protection for the anterior-most neuroectoderm from the posteriorizing activity of WNT.

Human brain organoids show as well a lack of a reproducible topographic organization in forebrain. A recent study[16] shows that triggering SHH protein gradient in developing forebrain organoids leads to formation of major forebrain subdivisions positioned with a *vivo*-like topography.

Consistent with this observation, we predict that engineering aggregates of ESC secreting WNT (or WNT and NODAL) antagonists such as *Dkk1*, *Cer1* and *Lefty1*, when merged anteriorly at early embryoid stages (D3–D4) at the opposite end of the original signalling centre may modulate NODAL and/or WNT gradient of activity, allowing the formation and differentiation of a patterned forebrain. Future studies will address this point. Because our strategy allows for the control of the time of merging, size of the aggregates used as a signalling centre and the level of expression of morphogens and morphogen antagonists, we are confident that we will be able to improve the formation and patterning of the anterior-most embryonic tissues in our embryoids.

The second limitation of our in vitro model is that, while the AP axis is easily defined by the position of the signalling centre secreting NODAL and WNT, the DV axis appears stochastically. While the frequency with which we get a well-built DV axis is robust, this is not experimentally controlled. In consequence, we find lower frequency in generating lateral/ventral tissues (intermediate mesoderm, surface ectoderm) than dorsal and dorsal-lateral tissues. In zebrafish, we have been able to circumvent this limitation by experimentally generating perpendicular gradients of Nodal and BMP[22].

Finally, to limit any interference of the culture media with the development of embryoids, we intentionally grow them in minimal conditions. Reports using gastruloids and TLS models[13,14] have shown that morphogenesis can be improved by growing aggregates in Matrigel. Future studies will explore the potential for this strategy to promote further development of the embryoids in time.

By experimentally engineering an asymmetric morphogen signalling centre acting as an organizer, we have generated a neurula-stage embryo-like (embryoid) entity that provides a powerful and promising model for further in vitro studies. This model holds the potential for further manipulating gradients, modelling diseases, performing drug screening and even for development of a human counterpart.

## Methods

**ESC lines**. Embryoids were generated from various ESC lines derived from the mouse strain 129/Ola. This includes the *Bra-GFP*[69] that is a *brachyury/EGFP* knock-in ESC line, the E14TG2a line (non-labelled, wild-type cells, ATCC CRL-1821), the 129 ESC with *GFP* line (which expresses *GFP* constitutively, Cyagen MUAES-01101) and the *Sox1-GFP*[28] that is a *Sox1/EGFP* knock-in ESC line. All these lines have been used either alone or in combination to generate embryoids described in this study.

**ESC culture**. Mouse ESC were grown on 0.1% gelatine-coated plates at 37 °C and 5% $CO_2$, in a serum-free medium made of KnockOut Dulbecco's modified Eagle medium (KO-DMEM, Gibco 10829018), supplemented with 15% KnockOut Serum Replacement (KSR, Gibco 10828028), 2 mM L-Glutamine (Gibco 25030081), 1 mM non-essential amino acids (NEM NEAA, Gibco 11140076), 100 μM β-mercaptoethanol (Gibco 21985023) and 1x Antibiotic-Antimycotic (Gibco 15240062). They were maintained in a pluripotent state by using ESGRO Leukemia Inhibitory Factor (LIF 1 μM, Millipore ESG1106) and two inhibitors (called 2i), PD0325901 (MEK inhibitor 1 μM, Selleckchem S1036) and CHIR99021 (GSK-3 inhibitor 3 μM, Selleckchem S2924) to block the MAPK/ERK and glycogen synthase kinase 3β (GSK-3β) pathways.

**Exit from the pluripotency state and formation of embryoids**. ESC were dissociated by Accutase (Sigma-Aldrich, A6964) treatment for 2 min at 37 °C. Single-cell suspension was produced by trituration of cell clumps. Cells were then pelleted by centrifugation at $200 \times g$ for 5 min and resuspended in serum-free medium with KSR and LIF but not 2i. After Trypan Blue labelling, living cells were counted and diluted to the desired concentration (10 cells/μl) in the same medium. Using this cell suspension aggregates of 50 or 100 cells (drops of 5 and 10 μl, respectively) were made by the hanging drop method[70] for 18–20 h at 37 °C in the presence of 5% $CO_2$. We define Day 0 (D0) as the time when the drops are placed on the lid of the cell culture plate.

After their formation, aggregates were collected, transferred into a neutral medium, N2B27 (DMEM/F12 Gibco 11330032, Neurobasal Medium Gibco 21103049, N2 Supplement Gibco 17502048, B27 without Vitamin A Gibco 12587010, β-Mercaptoethanol Gibco 21985023, Antibiotic-Antimycotic Gibco 15240062), and cultured in hanging drops at 37 °C 5% $CO_2$ for 2 additional days during which cells of the aggregates progressively exit the pluripotency state.

At Day 2.6, the small aggregates made from aggregation of 50 cells at D0 were collected, placed in another drop of N2B27 containing 10 μg/ml BMP4 protein (R&D System P21275) and incubated for hours at 37 °C in presence of 5% $CO_2$. At D3, aggregates incubated with BMP4 were collected and washed in N2B27 to remove traces of BMP4. Then, they were placed individually in a well of a round bottom ultra-low attachment 96-well plate (Nucleon Sphera Microplates, Thermo Scientific 174929) filled with 100 μl of N2B27 and containing one naive aggregate made at D0 by aggregation of 100 cells. After about 1 h, the two aggregates in contact at the bottom of the well merged together into a single entity that we named an embryoid. These embryoids were cultured in the same plate, in N2B27 medium, at 37 °C, 5% $CO_2$ for up to 6 days.

**PCR analysis**. Between 96 and 288 embryoids were collected according to their stage of development, rinsed in PBS and total RNA was extracted using the RNeasy Mini kit (Qiagen, Cat No./ID: 74104). Tissue disruption and homogenization were performed with Qiagen buffer RLT containing β-mercaptoethanol by gentle pipetting. First-strand cDNA was synthesized from 1 μg of total RNA, using 50 μM oligo-dT primer (Invitrogen) and 200 U/μl M-MuLV Reverse Transcriptase (New England BioLabs). Amplification using primers listed in Supplementary Table 4 was done using a denaturation step at 95 °C for 10 min, followed by 35 cycles of denaturation at 95 °C for 1 min, primer annealing at 58 °C for 30 s and primer extension at 72 °C for 30 s. Upon completion of the cycling steps, a final extension at 72 °C for 5 min was performed and the final reaction product stored at 4 °C. The experiments were carried out in duplicates for each data point.

**In situ hybridization and imaging**. Embryoids were collected at different developmental stages, fixed overnight at 4 °C in 4% paraformaldehyde (PFA) and stored in methanol at −20 °C. In situ hybridization was performed as previously described[71]. Antisense RNA probes were made using either digoxygenin-11-UTP or fluorescein-12-UTP using linearized DNA or PCR amplicons as templates. Double colour in situ were performed by using two antisense RNA probes labelled with different haptens (digoxygenin or fluorescein). Presence of hybridized RNAs was revealed using alkaline phosphatase conjugated antibodies and NBT/BCIP or Fast Red substrates. As a control for specificity, each probe used was tested first on mouse embryos (gifts from Dr. X. Lu, Dr. M. Sequeira-Lopez and Dr. A. Sutherland) at stages for which the corresponding gene was known to be expressed. List of probes, sequences, method for the synthesis of the antisense RNA probes are provided in Supplementary Data 2 and the expression territories of the corresponding genes in the mouse embryo in Supplementary Table 5.

Representative images were acquired using a Leica macroscope M420 or a Compound microscope with differential interference contrast (DIC) Leica DMRA2 both equipped with a digital camera (Roper Scientific, Coolsnap CCD) and the coolsnap software v1.2. For each picture, the background surrounding the

embryoids has been homogenized to white. Fibres, particles, bubbles present in the images of the documented embryoids were removed using Photoshop. No alterations of the biological data of the embryoids have been made. Contrast and brightness have been slightly adjusted. Original untreated images are available upon reasonable requests to the corresponding author.

**Cryosections**. Embryoids were fixed overnight at 4 °C in 4% paraformaldehyde in 1XPBS, incubated in 30% sucrose overnight, then for 30 min in a 1:1 mix of Tissue-Tek O.C.T. compound. After three washes with O.C.T. compound, embryoids were then snap-frozen in O.C.T. medium and placed at −80 °C for cryosectioning. Sections of 10 μm were made using cryostat and placed on Superfrost Plus Microscope Slides.

**Immunohistochemistry and imaging**. Embryoids were fixed overnight at 4 °C in 4% paraformaldehyde in PBS then washed three times for 20 min in PBS containing 1% Triton X100 (Sigma-Aldrich, 93418). Embryoids were incubated for 1 h at room temperature in a blocking buffer: PBS-1%Triton, 10% Sheep Serum (MP Biomedicals, SKU 08642951), 1% BSA (Bovine Serum Albumin Fraction V, Sigma A3059) then incubated overnight at 4 °C in a medium containing the primary antibody. Primary antibodies used included: anti-Acetylated Tubulin (Sigma-Aldrich T7451) used at 1:400 dilution in the blocking buffer, anti-Laminin (Sigma L9393) used at 1:300, anti-Phospho-Histone H3 (Cell Signaling Technology 9701) used at 1:200, anti-Tubulin β3 (BioLegend 801201) used at 1:750 and anti-doublecortin (Abcam, ab18723) used at 1:200; anti-BRA (Abcam, ab209665) used at 1:1000; anti-GFP (Novus Biologicals, NB600-308) used at 1:400; anti-GATA6 (Invitrogen, PA1-104) used at 1:400; anti-NANOG (Invitrogen, 14-5761-80) used at 1:500; anti-Sox2 (Millipore, ab5603) used at 1:500 and anti-Sox17 (R&D Systems, AF1924) used at 1:20. Embryos were then washed three times for 20 min with PBS-1%Triton and incubated 2 h at room temperature in a medium containing the secondary antibody: goat anti-rabbit Alexa Fluor 488 (Thermofisher, A11008) or a goat anti-mouse Alexa Fluor 546 (Thermofisher, A11030) used at a 1:800 dilution in the blocking buffer. After three final washes of 20 min in PBS-1%Triton, embryoids were mounted in Vectashield mounting medium (Vector, H-1000). Cell nuclei were labelled by incubation with Hoechst 33342 (Invitrogen, H3570) at a 1/200 dilution.

For actin cytoskeleton staining, embryoids were incubated for 1 h in Alexa Fluor 546 Phalloidin (ThermoFischer, A22283) used at 1:200 diluted in PBS-1% Triton. Representative images were acquired using a Leica TCS LSI confocal macroscope and the LAS AF (Leica Application Suite Advanced Fluorescence) software.

**Statistics and reproducibility**. Number of embryoids analysed by in situ hybridization or immunolabelling and number of independent experiments performed are presented in Table 1.

**Video microscopy**. Embryoids were mounted in 4% agarose wells (Sigma A9539) on 35-mm glass-bottom dishes (MatTek Corporation P35G-0.170-14-C) covered with N2B27 and maintained in a chamber at 37 °C, 5% $CO_2$ for the whole length of acquisition. Imaging was performed using a Leica DMi8 motorized fluorescence microscope with LED light source (Leica 15024) with a Leica-DFC9000GTC-VSC10184 camera and the Leica application suite X (LAS X).

**Bulk transcriptome analysis**. Total RNA was isolated from 10 pooled embryoids collected at D7 and D8 using the RNeasy Micro Kit (Qiagen, 74004). RNA-seq libraries were constructed by the University of Virginia, Department of Biology Genomic Core Facility Libraries using a NEBNext Ultra DNA Library Prep Kit for Illumina (New England Biolabs, E7370S), with standard TruSeq-type adapters. The size and concentration of the libraries were assessed using an Agilent 2100 Bioanalyzer. Single-end sequencing of 75-bp was performed by the Genome Analysis and Technology Core of the University of Virginia School of Medicine, on an Illumina NextSeq500 sequencing system. Sequencing reads were filtered and mapped to the mouse genome build mm10 using the STAR alignment programme (v2.6.1a). The expression values of each gene were normalized using TPM. PCA and DEGs were calculated using the R package DESeq2 (v1.22.2). DEGs were deemed significant if they passed the following cut-off parameters: TPM > 1, absolute value of log2 fold change >1, and Q-value (adjusted $p$ value) <0.05. Gene ontology (GO) analysis was performed using Fisher's exact test, and the false discovery rate (FDR) was controlled by the BH method. Heatmaps were generated using the R package pheatmap (v1.0.12).

**Single-cell RNA analysis**

*Embryoids dissociation*. 90 D8 embryoids were pooled and dissociated with Accutase (Gibco: #A11105-01) for 30 min at 37 °C with pipetting every at 5-min intervals. Cells were filtered using 40 μm Flowmi cell strainers (Belart Cat number 136800040). Cells were then fixed in prechilled methanol for 30 min on ice and stored at −20 °C. Cells were rehydrated in 1xDPBS −1.0%BSA −0.5 U/μl RNase inhibitors and filtered again to eliminate debris and cell agglomerates. Cells were immediately counted using the Countess 3 automated cell counter (Thermofisher Scientific) and processed for 10x Genomics scRNAseq.

**Table 1 Number of embryoids analysed by in situ hybridization or immunolabelling and number of independent experiments performed for Figs. 1, 2, 3, 4, 5 and 6.**

| Panel | Number of embryoids analysed | Number of independent experiments |
|---|---|---|
| Figure 1 | | |
| b | 5154 | 11 |
| d | 35 | 2 |
| e | 26 | 2 |
| f | 91 | 3 |
| i | 12 | 2 |
| j | 15 | 2 |
| k | 60 | 5 |
| l | 137 | 2 |
| m | 46 | 3 |
| n | 58 | 2 |
| p | 67 | 2 |
| q | 63 | 2 |
| Figure 2 | | |
| b | D3.5: 12 | 2 |
| | D4: 16 | 2 |
| | D4.5: 21 | 2 |
| e | D3.5: 18 | 2 |
| | D4: 13 | 2 |
| | D4.5: 19 | 2 |
| g | D3.25: 65 | 5 |
| | D3.5: 18 | 2 |
| | D4: 52 | 4 |
| | D4.25: 23 | 2 |
| | D4.5: 18 | 2 |
| | D5: 19 | 2 |
| h | D3.5: 23 | 2 |
| | D4.5: 18 | 2 |
| | D5: 21 | 1 |
| l | D4: 17 | 1 |
| | D4.5: 16 | 1 |
| | D5.5: 23 | 1 |
| Figure 3 | | |
| a | 10 | 1 |
| b | 13 | 1 |
| c | 5 | 1 |
| d | 42 | 5 |
| e | 28 | 2 |
| f | 16 | 1 |
| g | 101 | 3 |
| h | 47 | 2 |
| i, j | 7 | 2 |
| k | 14 | 4 |
| l | 100 | 6 |
| m | 15 | 2 |
| n | 10 | 1 |
| o, p | 18 | 2 |
| q | 104 | 5 |
| r | 59 | 3 |
| s | 51 | 3 |
| u | 12 | 2 |
| v | 106 | 3 |
| Figure 4 | | |
| a | 153 | 6 |
| b | 42 | 2 |
| c | 53 | 2 |
| d | 69 | 4 |
| e | 30 | 2 |
| f | 12 | 2 |
| g | 47 | 3 |
| h | 26 | 1 |

**Table 1 (continued)**

| Panel | Number of embryoids analysed | Number of independent experiments |
|---|---|---|
| i | 70 | 4 |
| j, k | 63 | 5 |
| l, m | 181 | 11 |
| n | 40 | 2 |
| o | 44 | 2 |
| p | 17 | 1 |
| q | 36 | 2 |
| r–t | 150 | 9 |
| Figure 5 | | |
| a, b | 87 | 3 |
| c | 21 | 2 |
| d | 43 | 2 |
| e | 88 | 3 |
| f | 58 | 2 |
| g | 89 | 4 |
| h | 29 | 1 |
| i | 39 | 1 |
| j | 29 | 1 |
| Figure 6 | | |
| e | 44 | 3 |
| f | 16 | 1 |
| g | 46 | 2 |
| i | 82 | 3 |

The variability of expression pattern for each gene is described in Supplementary Fig. 4 for data of Fig. 1; Supplementary Fig. 7 for data of Fig. 2; Supplementary Figs. 9 and 10 for data of Fig. 3, Supplementary Fig. 12 for data of Fig. 4; Supplementary Fig. 13 for data of Fig. 5 and Supplementary Fig. 14 for data of Fig. 6.

*10x Genomics scRNAseq*. Cells were processed by the UVA Genome Analysis and Technology Core (RRID SCR_018883) using the 10x Genomics Chromium Single Cell 3′ (v.3 Chemistry) gene-expression kit (10x Genomics Cat. number 1000269) targeting a cell recovery up to 10,000 sequenced cells according to Manufacturer's recommendations. Sequencing was then performed on the Illumina next-seq sequencing platform.

*10x Data pre-processing and comparison with mouse*. scRNAseq dataset was processed with Cell Ranger 3.1.0 to demultiplex the raw base call files, generate the fastq files, perform the alignment against the mouse reference genome mm10, filter the alignment and count barcodes and UMIs. Quality control and cluster determination were performed using the Seurat package (V3.2.1) in Rstudio (V1.3.1093). Single-cell data were loaded with a minimum requirement of 200 features and 3 cells. Cells with low library complexity (under 1700 genes and mitochondrial-fraction >5% were excluded[72]. Identification of cell clusters was performed by comparison to a previously published set of mouse embryo (E6.5 to E8.5) sc-transcriptomes[65].

The mouse dataset was filtered to only include only the relevant time point (E8.5). Expression data were independently normalized and variable features were detected. Integration anchors were identified and an integrated analysis was then performed on the embryoid and mouse cell datasets to identify common cell types. PCA was performed to calculate a joint UMAP. Generation of clusters was performed following standard workflow steps (FindNeighbors, Findclusters). Finally, one small cluster was removed likely due to the presence of stressed cells (high mitochondrial RNA counts and low total RNA counts).

*Pseudotime analysis*. Pseudotime analysis was performed using the R Package Monocle 3. UMAP and clusters identification previously performed for the embryoids dataset were used to analyse trajectories for pseudotime.

**Reporting summary**. Further information on research design is available in the Nature Research Reporting Summary linked to this article.

## Data availability

The source data underlying Figures are provided as Supplementary Figures describing the number of experiments, the number in biological samples analysed and a description of variability of gene-expression pattern. Bulk and scRNAseq data have been deposited in the National Center for Biotechnology Information Gene Expression Omnibus database under accession code GSE142309 for Bulk-RNA sequencing and GSE168539 for the single-cell RNA sequencing. Source data are provided with this paper.

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

## Acknowledgements

We would like to thank Abigail Antoine and Jinwoo Lee for their contribution in the early phase of the project. We thank Dr. R. Keller, Dr. A. Sutherland and Dr. R. Bloodgood for comments on the manuscript. We thank Dr. G. Keller and Dr. A. Smith for the Bra-GFP and Sox1-GFP ESC lines, respectively, Dr. J. Yu for Pax6 cDNA, Dr. A. Sutherland, Dr. X. Lu, Dr. A. Lantin Malt and Dr. M. Sequeira-Lopez for fixed mouse embryos and Dr. K. Hirschi for access to her Leica DMI8 microscope. The work was supported by March of Dimes (1-FY15-298 to B.T.), the Jefferson Trust (FAAJ3199 to C.T.) and the University of Virginia (B.T. and C.T.). M.O.-M. was supported by CNPq-Brazil (200535/2014-5).

## Author contributions

P.-F.X., R.M.B., J.F., M.O.-M. and T.C. designed, performed and interpreted biological experiments. B.T. and C.T. conceived the project, oversaw its implementation and wrote the manuscript.

## Competing interests

The authors declare no competing interests.
