## [Peer Review File · Nature Communications]

Reviewers' Comments:

Reviewer #1:

Remarks to the Author:

In this manuscript, Xu et al describes a method to create a neurula-like stage mouse embryoid by using two differently cultured mESC aggregates (one small aggregate cultured with BMP4 and another larger "naïve" aggregate) and having them fuse to form the embryoid. The BMP4 cultured aggregate acts like a signaling center/organizer to control more embryonic-like development. By utilizing this method, the authors claim to create a neurula-like mouse embryo model. They mainly utilize IHC and bulk RNA transcriptomics data as evidence for their claims. With this process, they observe an elongation behavior like the AP elongation of the embryo. They also found endoderm gut expression in a similar order as known in vivo. They found that the embryoids exhibited some neural tube-like structures based on some staining and polarity data.

First and foremost, this manuscript language-wise requires much revision. Besides grammar errors and typos, there are many areas where the manuscript makes unjustified overstatements. For example, Lines 130-132, it is not valid to claim the structures formed are equivalent to the notochord. It can be notochord-like. You can say it resembles the primate streak and not that it is the primitive streak. This occurs throughout the paper and must be revised (e.g. Line 184, 240). These are only a few examples; there are many, many such unsubstantiated overstatements throughout the entire manuscript.

Starting from their abstract, I have an issue with their claim about "remarkably well patterned". That is extremely vague and I do not think they provide much evidence for this claim in the paper. There is much more staining necessary to claim true patterning. It is one thing to have the transcriptome show that many of the genes that are patterned are in the structures, but it is entirely a different thing to see if the cells expressing those genes are in the correct place.

This paper I think struggles a bit with novelty and actual impact. Currently, the main issues faced is that embryoids/gastruloids do not express many anterior portions of the embryo. This manuscript/method faces the same issue. Morphogenetically, this paper does look a bit better. In the discussion they do talk about how this method allows for them to control the time at which the signaling center is used, the size of the signaling center, and the intensity. I agree that could be true, however they do not provide enough evidence to characterize and support the advantages of this. I especially have issue with the claim about intensity of the signaling center. Are not the size and intensity dependent on each other? Overall, I am not sure if this paper is providing much in novelty. The model does have some more regions closer to the anterior end which does help the paper. They mention in the intro that the other models have mostly posterior and lack an anterior ("anterior truncation" as they call it).

Linked to the "well patterned" comment. I think the paper needs more staining and detailed molecular characterization and authentication. Like in 182 is there no hindgut staining that could be done? I am not convinced by the neurulation portion of this manuscript, cell shape, polarity, and some staining is not sufficient to claim a neural tube. I believe the authors should conduct more DV and AP staining to see actual patterning in the neural tube-like structure. There are many established molecular markers for DV and AP patterning of the neural tube.

All in all, this is an interesting paper, representing some useful improvements over the existing mouse gastruloid model (including structures that resemble the primitive gut tube, the neural tube, and primitive cardiac tissues). But there are very obvious major weaknesses, including a lack of careful and detailed molecular characterization to support their statements and a lack of quantitative discussions of their findings. The authors will have to conduct single-cell RNA-seq to characterize the cell lineages in their model and compare the data with in vivo mouse data at comparable developmental stages.

Some other comments are listed below.

1. Figure 1. I would like to see a schematic figure showing how the aggregates are created and how they are different (timeline of culturing, media used).
2. Figure 5. I think it would be helpful labeling orientation of the embryoids what is DV and AP.
3. Figure 6. What about comparing with other data? Is there any other data to compare with to confirm? I think that could be more informative.
4. Figure 8. If they had staining, I would believe the patterning claim more.
5. Figure 1 Supp. I would add a fitted line and equation.
6. I would like to see a better characterization of the BMP-4 treated aggregate before it merges with the ESC aggregate. This could include immunostaining and scRNA-seq to confirm the identity of the cells that make up the aggregate.
7. For figure 1 I would have liked to see a cartoon/schematic of the protocol as well as quantification of their results. Did 100% of the aggregates successfully merge? Was the localization of these markers identical in all of their experiments, or was there a subset of the structures that displayed different marker localization?
8. For figure 2, why use only 10% of the cells tagged with GFP? It is definitely better for tracking movement of cells because there is less noise, but they could have done at least one experiment in which they used 100% BRA-GFP cells to look at localization. Using only 10% could lead to misleading results.
9. For figure 2 I would have liked to see better characterization of temporal dynamics of BRA localization. Where are the quantifications? Could they quantify BRA localization at intermediate time points to get a better idea of temporal dynamics? Additionally, could they show immunostaining results for BRA to see if the BRA-GFP cells accurately represent what is happening in their system?
10. Where are the quantifications for figures 3 and 4? I would have liked to see more careful characterization of temporal dynamics of marker localization as well as a better characterization of cavity development in figure 3. They use these figures to show the formation of mesoderm and endoderm derivatives, but this is not enough. Where is the scRNA-seq data? They should compare it with mouse embryo mesoderm and endoderm derivatives to show how similar their system is to the in vivo system.
11. Again, where are the quantifications for figure 5. What percentage of the structures develop the np? How does marker localization vary?
12. For transcriptome analysis, they compare embryoids with controls. Why not compare embryoids with mouse embryo data?

Reviewer #2:

Remarks to the Author:

The manuscript by Xu et al. proposes a novel method to generate an embryonic stem cells-based embryo model. The method relies on the fusion of two cell aggregates, one of which has been instructed through culture in presence of BMP4, resulting in the activation of the Wnt and Nodal pathways and hence providing a signalling center to the resulting aggregate. Remarkably, those "embryoids" are patterned in the anterior-posterior and dorsal-ventral axes, specify the three germ layers, and differentiate a large amount of specific populations, some of them organising as organ precursors such as the gut tube, the neural tube, the heart etc.

A number of protocols to generate embryo models from stem cells have been described those last years. This novel model, however, is the first to my knowledge to cover the post-gastrulation period with the generation of the three germ layers, in particular providing a platform to study neurulation in vitro within an embryo-like context. For that specific aspect, the authors might want to discuss how their approach complements the work of the Meinhardt lab regarding "neural tube in a dish" protocols.

One powerful aspect of the manuscript is the relative technical simplicity (including the absence of requirement for complex engineered substrate and/or microfluidics), and the very detailed description of the method, which suggests that the protocols might be quite easily reproduced and used by numerous teams. In addition, the source data provide useful information regarding the reproducibility of the protocol for each population.

One aspect that may be further explored is the path from "mesendoderm" to mesoderm and endoderm. The existence of a common progenitor in mouse is still a matter of debate (cf Probst et al. biorxiv). There is also recent work in gastruloids suggesting a mode of endoderm generation based on partial EMT/differential adhesion (Hashmi et al. and Vianello et al., both biorxiv). It would be interesting to test the presence of cells co-expressing *Bra* and/or *Mesp1* and *Sox17* and/or *FoxA2*, as well as evaluate the levels of E-cad in the prospective endoderm cells.

Overall, this is very elegant and relevant work, which has the potential to have an important impact in the developmental biology and neuroscience community. The data is of excellent technical quality, and the manuscript is super clear and complete, including the methods and statistics.

Reviewer #3:

Remarks to the Author:

In this manuscript, Xu et al introduce a new in vitro model for the early mouse embryo. They combine mESCs that have been pretreated with BMP4 with naïve mESCs and show that the pretreated embryos serve as a signaling center which causes patterning in the aggregate. The resulting structures show similarity to mouse embryo at late gastrulation and neurulation stages. Although several similar approaches have been published over the last few years, this is an interesting model with some novel aspects and could be particularly useful for probing inductive interactions between cells. I support publication but there are several issues that need to be addressed.

1. Statistics are insufficient throughout. For every observation, the authors need to note in the main text how many aggregates were analyzed and how many showed the expected phenotype. Statements that this information is available in supplementary data are not adequate.
2. Relatedly, information about reproducibility is lacking. The variability in size and shape parameters (e.g. length, width, volume etc) needs to be presented. Similarly, for each staining pattern, variability should be noted. Presumably not all aggregates look the same and the authors should accurately convey the spectrum of phenotypes.
3. It would be very helpful in understanding the model to know which tissues are contributed by the pretreated cells. Similar images to Figure 1a at later time points would be informative. Is the node-like structure derived from the graft? What about the notochord-like structure? Do both the pretreated and untreated cells contribute to all three germ layers?
4. The experiment in figure 2 requires quantification. The authors should plot the positions of GFP expressing cells along the axis of the embryoid.
5. In Figure 3, 5, the tube structures of the gut and neural tube are evident in the immunofluorescence but more difficult to observe in the in situ. However, the immunofluorescence images are not stained for fate markers so it is difficult to confirm the interpretation. It would be good to costain with markers of fate (e.g. Phalloidin/*Sox17* in figure 3, Phalloidin/*Pax6* in Figure 5).
6. In figure 4h, it is difficult to see the network of vessels the authors refer to. Better data should be provided or the claims softened.
7. The data in Figure 6 don't add much to that in Figure 7,8 and could be moved to supplementary material.
8. The statement in lines 69-71 is not entirely correct with regard to mammalian embryos. Not all signals come from the "embryo proper" but many are secreted from extraembryonic tissues. As, for example, the trophoblast is in contact with the edges of the cup-shaped embryo, it may

indeed provide signals particularly at the periphery.

9. The discussion is rather biased in comparing this model to others. The authors state that this model is more “controlled”, however, they do not define this word. In some ways, it is easier to control the precise concentration of an extracellular inducer than the concentrations of Nodal/Wnt which are induced in the pretreated cells, which will vary. It is also possible to withdraw extracellular inducers but the cell graft cannot be undone. The authors state that the fact that asymmetry arises without experimental control in other models is a drawback, however, I consider this a feature – symmetry breaking is one of the most important and interesting processes in mammalian development and the authors have bypassed this by experimentally breaking the symmetry. Overall, the authors should be more clear and precise about the advantages and disadvantages of their model compared to others.

10. The authors should revise claims about limited morphogenesis in gastruloids and priority claims about neural tube formation in vitro in light of recent publications and preprints – van den Brink et al Nature 2020, Veenvliet et al bioRxiv 2020.

11. The authors may wish to discuss Cederquist et al Nature Biotech 2019 which takes a similar approach to patterning in the nervous system.

Answers to Referee Comments

Reviewer#1

We would like to thank Referee 1 for his comments about our manuscript: ‘*this is an interesting paper, representing some useful improvements over the existing mouse gastruloid model (including structures that resemble the primitive gut tube, the neural tube, and primitive cardiac tissues)*’. We thank also Referee 1 for his/her useful suggestions to improve the paper. We are now able to provide a much stronger and more detailed manuscript. We looked at all comments in great details and did our best to address them all. Therefore, as requested, we made substantial revisions of our manuscript. We particularly focused on the following points:

A more detailed characterization of the structures in the embryoids

We used an extensive collection of specific markers (40 markers for *in situ* and/or immunolabeling) to provide a much more precise spatio-temporal description of the differentiation of the 3 germ layers in our embryoids (**Figure 1-6, Suppl Fig. 1-24**). We performed scRNA seq (**Fig. 8-10, Supplementary Fig. 19-23**) in addition of Bulk RNA (**Fig. 7, Supplementary Fig. 17-18**) and provide an extensive analysis for identifying the structures made in our embryoids (e.g.: clusters identity, pseudotime differentiation trajectory analysis).

Quantification/details of reproducibility

For each marker, we provide the number of experiments and of biological samples we analyzed as well as the variability in gene expression we observed, with images associated with variations from the main pattern and the percentage of embryoids associated to these variations (**Supplementary Figures 3, 4, 7, 9, 10, 12-14**).

Comparison with other published work

This is addressed throughout the manuscript when appropriate and in the discussion.

Detailed responses to Referee 1 (italics: Referee comments)

In this manuscript, Xu et al describes a method to create a neurula-like stage mouse embryoid by using two differently cultured mESC aggregates (one small aggregate cultured with BMP4 and another larger “naïve” aggregate) and having them fuse to form the embryoid. The BMP4 cultured aggregate acts like a signaling center/organizer to control more embryonic-like development. By utilizing this method, the authors claim to create a neurula-like mouse embryo model. They mainly utilize IHC and bulk RNA transcriptomics data as evidence for their claims. With this process, they observe an elongation behavior like the AP elongation of the embryo. They also found endoderm gut expression in a similar order as known in vivo. They found that the embryoids exhibited some neural tube-like structures based on some staining and polarity data.

First and foremost, this manuscript language-wise requires much revision.

The manuscript was read by colleagues. We switched to British English spelling.

Besides grammar errors and typos, there are many areas where the manuscript makes unjustified overstatements. For example, Lines 130-132, it is not valid to claim the structures

formed are equivalent to the notochord. It can be notochord-like. You can say it resembles the primate streak and not that it is the primitive streak. This occurs throughout the paper and must be revised (e.g. Line 184, 240). These are only a few examples; there are many, many such unsubstantiated overstatements throughout the entire manuscript.

This has been corrected all along the manuscript (e.g.: notochord to notochord-like, gastrulation-like, primitive-streak like, gut-like structure, likely foregut, likely hindgut, likely midgut, endocardial-like tubes, likely corresponding to the midbrain-hindbrain boundary, a flat epithelium likely corresponding to surface ectoderm).

Starting from their abstract, I have an issue with their claim about “remarkably well patterned”. That is extremely vague and I do not think they provide much evidence for this claim in the paper. There is much more staining necessary to claim true patterning. It is one thing to have the transcriptome show that many of the genes that are patterned are in the structures, but it is entirely a different thing to see if the cells expressing those genes are in the correct place.

As requested, we provide now a much more detailed spatio-temporal analysis of the structures made in our embryoids (40 markers for *in situ* and/or immunolabeling) to provide a much more precise spatio-temporal description of the differentiation of the 3 germ layers in our embryoids (**Figure 1-6, Suppl Fig. 1-24**).

We performed scRNA seq (**Fig. 8-10, Supplementary Fig. 19-23**) in addition to bulk RNAseq (**Fig. 7, Supplementary Fig. 17-18**) and provide an extensive analysis for identifying the structures made in our embryoids. For example, for the neurectoderm, we have been able to reveal the presence of dorsal cell identity clustering together as well as ventral cell identity clustering together, correlating with their position (either dorsal or ventral) in the neural tube of the mouse embryo (within the cluster of spinal cord and midbrain-hindbrain).

In summary, the bulk and sc-transcriptomic analyses confirmed and extended, with a comprehensive list of markers, the results obtained by our analysis using *in situ* and immunolabeling and confirmed our observations that embryoids undergo extensive patterning in the three germ layers along AP and DV axes.

The transcriptomic data provided in addition of an extended analysis of gene expression pattern on whole embryoids establish that, not only these genes are expressed but that the cells expressing those genes show for a vast majority of them a correct spatio-temporal distribution.

We changed the wording in the abstract from ‘remarkably well patterned’ to ‘extensively patterned’ to report the best accurate possible our observations and avoid overstatements.

This paper I think struggles a bit with novelty and actual impact. Currently, the main issues faced is that embryoids/gastruloids do not express many anterior portions of the embryo. This manuscript/method faces the same issue. Morphogenetically, this paper does look a bit better.

Here are the novelties this embryoid model brings compared to other *in vitro* models:

Extensive development of the embryoids

We identified the formation of three germ layers with clear antero-posterior and dorso-ventral patterning (**Fig.1-6 labeling, Fig 7-10: transcriptomics**).

From D3-D7, we were able to document the presence of an endodermal layer and its development starting with the formation of a primitive endoderm-like cell population, then the

formation of a visceral endoderm epithelium and the formation of an endoderm epithelium surrounding a lumen (resulting from the merging of multiple cavities surrounded by a *Sox17* expressing epithelium). This endoderm epithelium is made of a mosaic of visceral and definitive endodermal cells as it has been reported for the endoderm epithelium of the mouse embryo. It differentiates into a primitive gut tube that folds into an anterior pocket (likely foregut and expressing markers specific of the mouse embryo foregut diverticulum) followed on the dorsal side by a straight epithelium (likely midgut and expressing *Nepn*, a marker of the midgut in the mouse embryo), connecting a posterior pocket (likely hindgut that express specific markers of the hindgut diverticulum). On the ventral side, in the continuity of the hindgut, the epithelium extends and often exits the embryoids forming an external loop that may represent a rudimentary presumptive yolk sac and the territory connecting it to the primitive gut tube (to the hindgut posteriorly and the ventral foregut anteriorly) may be an equivalent of a rudimentary yolk stalk (**Fig. 3, 7-8; Suppl Movie 1, Suppl Fig. 9-11**).

Embryoids show clear evidence of a gastrulation process during which mesodermal cell movements appear similar to those observed in the mouse embryo, with a primitive streak-like domain located at the posterior tip from where cells migrate anterior to form the mesoderm germ layer. (**Fig. 2, Suppl Fig. 5 - 7**)

Embryoids also differentiate mesodermal derivatives: notochord-like and node-like structures, segmented paraxial mesoderm and intermediate mesoderm-like domains, anterior lateral plate mesoderm-like derivatives (beating heart/cardiac tissue, primitive erythroid cells, vasculature, reticular network of blood vessels formed both by vasculogenesis and angiogenesis) (**Fig. 4, 7-9; Suppl Fig. 12**)

We identified in our embryoids an ectodermal layer in which a neural plate differentiates and can fold into a neural tube. Embryoids display much more than a truncal-like region, they develop from an isthmus-like region at the midbrain-hindbrain boundary (expression of *En2* and *Fgf8* (**Fig. 5e-f**), a hindbrain expressing *Egr2* (confirmed with expression of *Cyp26b1* in scRNAseq) and a spinal cord elongated down to the tip of the tail. The AP patterning of the neural plate has been shown using anterior markers (*En2*, *Fgf8*, *Egr2*) as well as the transcription factors *Hoxd4*, *Cdx2* and *Hoxd9*. For these last genes we quantified the extent of their expression pattern in the neural tissues from posterior to anterior and demonstrated that their expression along the AP axis of the embryoids is similar to their expression in the mouse embryo (**Suppl Fig. 13**).

At D8, a front view of an embryoid reveals a folded neural plate that starts to close into a neural tube and for which dorso-lateral hinge points are visible (**Fig. 6d**). The neural plate, expressing the neural marker *Sox2* (both by *in situ* hybridization and immunolabeling) exhibits antero-posterior and dorso-ventral patterning (e.g.: expression of *Olig2*, a marker of progenitors of motor-neurons and oligodendrocytes) (**Fig. 6f**) is seen ventrally immediately above a *Shh* expressing domain and expression of the neural crest cell marker *Sox10* (**Fig. 6g-h**) is seen dorsally). These data were confirmed by scRNA seq (**Fig. 10**) in which we show that the dorsal neural identity of marker genes for the dorsal or the ventral neural tube of the mouse embryo cluster together, according to their expression along the DV axis of the neural tube of the mouse embryo.

Morphology of the embryoid neural plate revealed that it is made of a monolayered pseudo-stratified columnar epithelium with individual neuroectodermal cells extending from its apical to basal surface (**Fig. 6j**). This neuroepithelium shows an apical-basal polarity with apical actin filaments and presence of an extracellular matrix containing laminin at its basal side (**Fig. 6k**). Similar to a typical vertebrate neuroepithelium, in the embryoids, mitosis occurs at or near the

apical surface (Fig. 6l) while cell bodies of *TUBB3* expressing neurons are found basally (Fig. 6m). At D8, neurons extend their axons both within the developing neural plate/tube and toward the non-neural part of the growing embryoids (Fig. 6n-o) and at D9, in the anterior part of the neural plate, *TUBB3* expressing neurons appear in multiple cell layers (Fig. 6p) reminiscent of those observed at the onset of cortical neurogenesis in the hindbrain of the mouse embryo.

Altogether, in the embryoids, the organization of structures (from the midbrain-hindbrain boundary to the tip of the tail) is highly similar to that of a typical vertebrate embryo, organized around an axial mesoderm with a notochord-like domain below a neural plate and above an endoderm epithelium, with AP and DV patterning typical of a mouse embryo around mid-gestation.

In the discussion they do talk about how this method allows for them to control the time at which the signaling center is used, the size of the signaling center, and the intensity. I agree that could be true, however they do not provide enough evidence to characterize and support the advantages of this. I especially have issue with the claim about intensity of the signaling center. Are not the size and intensity dependent on each other?

In our protocol, aggregates of two different sizes (50 and 100 cells) are generated at Day 0 (D0), then cultured for three days in a basal medium devoid of factors maintaining cell pluripotency. On the third day (D3), small aggregates (initially made of 50 cells) are incubated for 8 hrs. in presence of purified mouse BMP4 protein. These BMP4 treated (instructed) aggregates are then individually placed at the bottom of wells of ultra-low attachment plates in contact of individual large aggregates, initially made of 100 cells and that are not exposed to BMP4 stimulation (untreated aggregates). Within one hour, instructed and untreated aggregates merged spontaneously into a larger structure we call embryoid. All the parameters described above (size of aggregates, BMP4 concentration, duration of culture, media, duration of instruction) have been determined prior to the present study. The current focus of this study was to analyze the outcome on the development of embryonic structures, using labeling techniques and genomics.

The method we use brings a number of advantages such as the possibility of changing the size of the organizer, mix different cell lines and potentially create other signaling centers that may secrete other morphogens or morphogen antagonists.

The analysis of the different parameters that can be changed experimentally was not the purpose of the study we present. However, we provide an example in **Suppl Fig. 24**, showing the effects of increasing BMP dosage for instruction. This favors formation of the heart, blood and gut-like structures, by both increasing the frequency of embryoids displaying these tissues as well as increasing their size (particularly for heart and gut tube).

Overall, I am not sure if this paper is providing much in novelty. The model does have some more regions closer to the anterior end which does help the paper. They mention in the intro that the other models have mostly posterior and lack an anterior (“anteriortruncation” as they call it). Linked to the “well patterned” comment. I think the paper needs more staining and detailed molecular characterization and authentication.

Addressed above

Like in 182 is there no hindgut staining that could be done?

We always have been very careful in choosing markers that are specific to one territory (when possible) to make sure we identify unambiguously the structure we see. They are not that many that only labels a single tissue. For identifying hindgut in an unambiguous manner we now use the hindgut diverticulum *Rnfl28* as well as *Apela*, which is expressed in the ventral foregut and in the hindgut of the mouse embryo. Expression of these markers is shown in **Fig. 3r** and **Fig. 3s**.

I am not convinced by the neurulation portion of this manuscript, cell shape, polarity, and some staining is not sufficient to claim a neural tube. I believe the authors should conduct more DV and AP staining to see actual patterning in the neural tube-like structure. There are many established molecular markers for DV and AP patterning of the neural tube.

We revised our analysis of the neural plate/tube in the embryoids:

Using an extensive collection of *in situ*/immunolabeling markers

Again, we tried to use markers as specific as possible to identify unambiguously the structures we see. We now describe the AP and DV patterning of the neural tube in two figures (**Fig. 5-6**) instead of one.

In addition of showing expression of *Pax6*, *En2*, *Egr2*, *Oligo2*, *Sox10*, *Sox2*, we now provide pictures for expression of *Scube2* (hindbrain and spinal cord marker, **Fig. 5d**), *Fgf8* (a landmark of midbrain-hindbrain boundary/isthmus, **Fig. 5f**), *Hoxd4* (posterior hindbrain, rhombomeres 6/7 to the caudal region, **Fig. 5h**), *Cdx2* (posterior part of the spinal cord, **Fig. 5i**), *Nkx1-2* (caudal epiblast, **Fig. 5j**), *Hoxd9* (caudal end, **Fig. 5k**). We made cartoons presented on the right side of each *in situ* embryoid picture to show where these different markers are expressed in the mouse embryo at an equivalent developmental stage.

In Figure 6, we show morphologically folding of the neural plate and presence of dorso-lateral hinge points (**Fig. 6a, 6d**). *Sox2* expression (immunostaining and *in situ* hybridization) confirms its neuroectodermal nature (**Fig. 6b-c**). We added a picture of TUBB3 expression showing neurons extending their axons in the neural plate/tube and in the non-neural part of the embryoids (**Fig. 6o**). We show at D9, TUBB3 expressing neurons in multiple cell layers (**Fig. 6p**). We added a picture of a marker for early post-mitotic neurons (DCX), expressed in neurons extending their axons both along the neural tube and toward the non-neural part of the embryoids (**Fig. 6r**).

Perform a scRNA seq (Fig. 8-10) and compare the transcriptome of embryoids to the published transcriptome of mouse embryos at E8.5

This analysis (**Fig. 8-10, Supplementary Fig. 19-23**) reveals the presence along the posterior to anterior axis of the neuromesodermal progenitors in the caudal part followed anteriorly by the spinal cord, and a midbrain-hindbrain domain, a floor plate domain and even a small cluster of cells with forebrain identity. This establishes clearly and confirms the observations made using molecular markers (*in situ* hybridization and immunolabeling). In addition, looking at the clustering of cells specific of dorsal and ventral cell identities as defined by the expression pattern along the DV axis of the neural tube of the mouse embryo, we found that cells with dorsal identities clustered together, cells with ventral identity also clustered together in two clear domains of the spinal cord and midbrain-hindbrain cell clusters. In addition, cell expressing gene markers of intermediate position along DV axis of the neural tube cluster were in between dorsal and ventral clusters.

Altogether, scRNA transcriptome reveals clear AP and DV patterning in the neural plate/tube of our embryoids at D8. Together with *in situ* hybridization and histological

characteristics of the embryoid neural plate we are confident that embryoids display a neural plate/tube close to that of a mouse embryo.

All in all, this is an interesting paper, representing some useful improvements over the existing mouse gastruloid model (including structures that resemble the primitive gut tube, the neural tube, and primitive cardiac tissues). But there are very obvious major weaknesses, including a lack of careful and detailed molecular characterization to support their statements and a lack of quantitative discussions of their findings. The authors will have to conduct single-cell RNA-seq to characterize the cell lineages in their model and compare the data with in vivo mouse data at comparable developmental stages.

This is now addressed (extended collection of *in situ*/immunolabeling markers, variability, statistics, scRNA seq and comparison to controls and mouse embryo comparable developmental stages)

Figure 1. I would like to see a schematic figure showing how the aggregates are created and how they are different (timeline of culturing, media used).

We modified our schematics in Figure 1 (**Fig. 1a**) in order to detail the different steps of our protocol. This is also described in the Method section.

*Figure 5. I think it would be helpful labeling orientation of the embryoids what is DV and AP. This has been corrected in our new Fig. 5 (**Fig. 5a**).*

Figure 6. What about comparing with other data? Is there any other data to compare with to confirm? I think that could be more informative.

We now provide an analysis comparing bulk RNAseq of embryoids with controls (**Supplementary Fig. 17-18**) and mouse embryos of equivalent stages (**Fig. 7**).

Figure 8. If they had staining, I would believe the patterning claim more.

We haven't been clear enough in the description of the data presented in **Fig. 8** (previous submitted version). The only point we wanted to make was that embryoids lacked genes specific of the forebrain and midbrain but displayed all genes expressed in territories posterior to the midbrain-hindbrain boundary. The drawing on the right of the heatmap was just to indicate the territory of expression of the genes analyzed (showing that genes expressed posterior to the midbrain were expressed in the embryoids).

However, we now provide in our new **Fig. 5**, evidence of an AP patterning in the neural plate/tube with anterior genes: *En2* for the posterior midbrain, *Fgf8* for the midbrain-hindbrain boundary, *Egr2* for the hindbrain and a quantified analysis of the expression pattern of *Hoxd4*, *Cdx2* and *Hoxd9* that are known to have their anterior most expression domain at different positions: posterior hindbrain for *Hoxd4*, middle of spinal cord for *Cdx2* and caudal spinal cord for *Hoxd9* (**Suppl Fig. 13**)

Figure 1 Supp. I would add a fitted line and equation.

This has been added in the figure (now **Supplementary Fig. 2**).

I would like to see a better characterization of the BMP-4 treated aggregate before it merges with the ESC aggregate. This could include immunostaining and scRNA-seq to confirm the identity of the cells that make up the aggregate.

We compared BMP4 treated aggregated to untreated aggregates at various times, before and after instruction, before and after merging and when they develop into embryoids: Control (untreated) at 0h, 8h, 24h, aggregates instructed with BMP4 at 0h, 8h, 24h, embryoids at D4 looking by RT-PCR for the expression of the following genes

Pluripotency: *Pou5f1*, *Nanog* (these genes are present in the control, present in the embryoids until early stages D4). NANOG has also been analyzed at D3 (soon after merging of both aggregate in double labeling with GATA6 in our study of the formation of the primitive endoderm)

Genes known to be induced in response to BMP4: *Wnt3*, *Nodal*

Downstream target (direct and indirect) of WNT3 and NODAL: *Bra*, *Eomes*, *Kdr*, *Sox17*, *Gata6* (stronger from 24h).

Altogether, there is very little happening in the aggregates at D2.66 (just before addition of BMP4) and we just observed the expected expression of genes that are induced in the embryo epiblast at the onset of gastrulation in response to BMP4 secreted by the extra-embryonic ectoderm. This is presented in **Supplementary Fig. 1**.

For figure 1 I would have liked to see a cartoon/schematic of the protocol as well as quantification of their results. Did 100% of the aggregates successfully merge? Was the localization of these markers identical in all of their experiments, or was there a subset of the structures that displayed different marker localization?

A schematic of the protocol is now shown in our new Figure 1 (**Fig. 1a**).

The statistic of successful merging of aggregates is now indicated in the text (N = 5154/5184 successful merging scored in 11 experiments with 99.29% success).

For each marker, we provide the number of experiments and of biological samples we analyzed as well as the variability in gene expression we observed with image associated with each variation from the main pattern and the percentage of embryoids associated to this variation ((**Supplementary Figures 3, 4, 7, 9, 10, 12-14**)).

For figure 2, why use only 10% of the cells tagged with GFP? It is definitely better for tracking movement of cells because there is less noise, but they could have done at least one experiment in which they used 100% BRA-GFP cells to look at localization.

Using only 10% could lead to misleading results.

This is correct, reducing the number of Bra-GFP cells (10%) allowed us to determine precisely the position for individual Bra expressing cells (**Fig. 2a**). We performed control experiments using 100% Bra-GFP cells in the instructing center that led to the same observations than with 10% cells, even we were unable to track them individually (now mentioned in the main text and shown in **Suppl Fig. 5**).

For figure 2 I would have liked to see better characterization of temporal dynamics of BRA localization.

Where are the quantifications?

Could they quantify BRA localization at intermediate time points to get a better idea of temporal dynamics?

Additionally, could they show immunostaining results for BRA to see if the BRA-GFP cells accurately represent what is happening in their system?

Quantification of cell position along the AP axis is now provided in **Supplementary Fig. 6** (with all raw data in **Supplementary Table 7**)

Localization over time of mesodermal cells was examined using double color *in situ* hybridization at various developmental stages (D3.25 to D5) in embryoids made by merging large naive aggregates made of *Bra-GFP* ESC to BMP4 instructed aggregates made of unlabeled ESC. We examined by *in situ* hybridization the expression of *Bra* (revealing all mesodermal cells induced by WNT3 and NODAL both in the signaling center and in adjacent cells of the naive aggregates - red) and of *GFP* (revealing cells from the naive aggregate expressing *Bra-GFP*).

We found that *Bra-GFP* expressing cells, revealed by the *GFP* RNA probe, formed an initial stripe located at around 40% from the posterior end of the embryoids. At D4-D4.25, GFP RNA containing cells were observed in a progressively more posterior position before they started to be detected (D4.5-D5) in the anterior part the embryoids (anterior to their initial position at 40% from the posterior end) (**Fig. 2g**). This observation supports the same migratory behavior suggested by the tracking of *Bra-GFP* labeled cell (**Fig. 2d-f**). These data further confirmed that the anterior most mesodermal cells at the onset of gastrulation first moved posteriorly, then after D4.5, moved anteriorly following what has been observed gastrulation of vertebrate embryos (easily visible in zebrafish where the most anterior mesodermal cells at the beginning of gastrulation, those further away from the margin, first move vegetally (while cell more vegetal in the margin at the onset of gastrulation have involuted and migrated toward the animal pole – similar to cells from the signaling center **Fig. 2a-c**). Then when the animal most mesodermal cells at the onset of gastrulation reach the involution point at the margin (like cells labelled in **Fig. 2 d-f**) they internalized and reverse direction migrating toward the anterior territories.

Altogether, tracking *Bra-GFP* fluorescent cells or following by *in situ* hybridization the position of cells from the stripe of mesodermal cells induced in the naive domain allowed us to reach the same conclusion and to claim that mesodermal cells of embryoids have a migrating behavior similar to vertebrate embryos (occurring in the mouse but not as easy to observe than in fish and frog). These data are described in the main text and were added to our new **Figure 2**.

This analysis has been completed by following overtime during gastrulation the expression pattern of two early migrating mesoderm markers *Eomes* and *Mesp1* (**Fig 2h, i**). Statistics and variability for *Bra/Bra GFP*, *Eomes* and *Mesp1* expression are presented in **Supplementary Fig. 7**.

Where are the quantifications for figures 3 and 4? I would have liked to see more careful characterization of temporal dynamics of marker localization as well as a better characterization of cavity development in figure 3. They use these figures to show the formation of mesoderm and endoderm derivatives, but this is not enough.

Statistics and variability (for **Figure 3** and **Figure 4**) are presented in **Supplementary Fig. 9, 10, 12** for the formation of the endoderm (**Fig. 3**) and for the analysis of mesoderm germ layer derivatives (**Fig. 4**).

Study of cavity development has been extended in **Figure 3** (*Sox17* expression in **Fig. 3h**; acetylated tubulin staining (**Fig. 3i-j**), SOX17 - Phalloidin labeling (**Fig. 3k**), alignment of the cavities at the midline and merged together into a single large cavity (**Supplementary Movie 1**, *Sox17* expressing endoderm epithelium in **Fig. 3l**).

The endodermal nature of the epithelium was confirmed at D7 by looking at additional gut markers including *Cldn4*, which is expressed in the whole gut epithelium of the mouse embryo at E8.5-E9.0 (**Fig. 3q**), *Apela* expressed in the mouse embryo in the ventral foregut as well as in the hindgut diverticulum (**Fig. 3r**) and *Rnf128*, a marker of the hindgut diverticulum (**Fig. 3s**) and the mosaic nature of the endoderm epithelium made of VE and DE cell is now reported in Fig. 3m-p.

Where is the scRNA-seq data? They should compare it with mouse embryo mesoderm and endoderm derivatives to show how similar their system is to the in vivo system.

scRNA seq has been performed and our data were compared to previously published scRNA-seq datasets of mouse embryos [E7.5, E7.75, E8.0, E8.25, E8.5 (Pijuan-Sala et al., *Nature*, 2019) and E9.5, E10.5 (Cao et al., *Nature*, 2019)]. This is presented in **Figures 8 to 10** and **Supplementary Fig. 19-23** and associated **Supplementary Tables**. This single-cell transcriptomic analysis confirmed and extended (with a comprehensive list of markers, pseudotime differentiation trajectory **Fig. 9-10**), our observation that embryoids undergo extensive patterning in the three germ layers along AP and DV axes.

Again, where are the quantifications for figure 5.

What percentage of the structures develop the np? How does marker localization vary?

We extended our characterization of the neural plate/tube and our data are presented in our new **Fig. 5** and an additional figure (**Fig. 6**). In about 15% of embryoids displaying a dorsal *Sox1-GFP(+)* epithelium, we observed the folding of the neural plate around D7 (**Fig. 6a**). Statistics and variability related to these figures and markers used are presented in **Supplementary Fig. 13-14**.

For transcriptome analysis, they compare embryoids with controls. Why not compare embryoids with mouse embryo data?

In addition to our previous comparison between embryoids and controls (now in **Supplementary Fig. 17-18**), we now provide comparison of bulk transcriptomes of D7 and D8 embryoids with the published bulk transcriptomes of mouse embryos at stages in between E8.0 and E9.0 (Wilson et al., *Nucl.Acids Res.* 2016 44(D1):D855-61) (**Fig. 7**).

Answers to Referee Comments
Reviewer#2
(italics: Referee comments)

We would like to thank Referee 2 for his comments about our manuscript and for acknowledging the novelties our study brings to the field of stem cells:

‘Remarkably, those "embryoids" are patterned in the anterior-posterior and dorsal-ventral axes, specify the three germ layers, and differentiate a large amount of specific populations, some of them organising as organ precursors such as the gut tube, the neural tube, the heart etc. This novel model, however, is the first to my knowledge to cover the post-gastrulation period with the generation of the three germ layers, in particular providing a platform to study neurulation in vitro within an embryo-like context.

We also thank Referee 2 for acknowledging that our protocol will be a useful resource to the scientific community:

‘One powerful aspect of the manuscript is the relative technical simplicity (including the absence of requirement for complex engineered substrate and/or microfluidics), and the very detailed description of the method, which suggests that the protocols might be quite easily reproduced and used by numerous teams’.

Finally, we thank Referee 2 for his overall evaluation of the work:

‘In addition, the source data provide useful information regarding the reproducibility of the protocol for each population. Overall, this is very elegant and relevant work, which has the potential to have an important impact in the developmental biology and neuroscience community. The data is of excellent technical quality, and the manuscript is super clear and complete, including the methods and statistics’.

Detailed responses to Referee 2

For that specific aspect, the authors might want to discuss how their approach complements the work of the Meinhardt lab regarding "neural tube in a dish" protocols.

This is now mentioned and discussed in the text.

One aspect that may be further explored is the path from "mesendoderm" to mesoderm and endoderm. The existence of a common progenitor in mouse is still a matter of debate (cf Probst et al. biorxiv). There is also recent work in gastruloids suggesting a mode of endoderm generation based on partial EMT/differential adhesion (Hashmi et al. and Vianello et al., both biorxiv). It would be interesting to test the presence of cells co-expressing Bra and/or Mesp1 and Sox17 and/or FoxA2, as well as evaluate the levels of E-cad in the prospective endoderm cells.

It has been recently reported that both in the mouse embryo and in gastruloids, definitive endoderm does not derive from mesendodermal progenitors that give rise to both endoderm and mesoderm but that the two lineages are already specified before ingression of mesoderm and endoderm cells at gastrulation (Probst et al., 2020). Cell-state transitions and collective cell

movement generate an endoderm-like region in gastruloids (A. Hashmi et al, bioRxiv, 2020, , Vianello and Lutolf bioRxiv, 2020).

Therefore, we investigated whether we could find evidence or not of a pool of mesendodermal progenitors in our embryoids. To do so, we performed double immunostaining at gastrula stage (D4) for the endodermal marker *Sox17* and the mesodermal marker *Bra* (**Supplementary Fig. 11**). We didn't observe any cells expressing both together the endoderm and the mesoderm markers, strongly supporting that in the embryoids, as observed in mouse embryo and in gastruloids, cells were already committed at time of gastrulation to endodermal or mesodermal fates rather than being bipotential, and representing a mesendodermal population.

We also evaluate expression of E-Cad in mesodermal and endodermal cells at early gastrula stage, looking for maintenance of E-Cad in endodermal cells and a decrease in E-Cad in Bra(+) cells. Even the experiments were repeated, the results we obtained were inconsistent and not clear enough to make a definitive statement. These E-Cad changes in mesodermal cells were not obvious in our embryoids and when looking in details at the image provided by Hashmi et al. we see E-Cad around Bra(+) cells that make the conclusion of the paper not completely convincing to us.

Answers to Referee Comments

Reviewer#3

(italics: Referee comments)

We would like first to thank Referee 3 for his comments about our manuscript and for acknowledging the novelties of our model and obtained results:

In this manuscript, Xu et al introduce a new in vitro model for the early mouse embryo. They combine mESCs that have been pretreated with BMP4 with naïve mESCs and show that the pretreated embryos serve as a signaling center which causes patterning in the aggregate. The resulting structures show similarity to mouse embryo at late gastrulation and neurulation stages. Although several similar approaches have been published over the last few years, this is an interesting model with some novel aspects and could be particularly useful for probing inductive interactions between cells’.

We thank Referee 3 for his overall evaluation of the work and support for publication.

We also thank Referee 3 for his/her useful suggestions to improve the paper. We are now able to provide a much stronger and more detailed manuscript. We examined all the comments in great details and did our best to address them all. We particularly focused on the following points:

A more detailed characterization of the structures in the embryoids

We used an extensive collection of specific markers (40 markers for *in situ* and/or immunolabeling) to provide a much more precise spatio-temporal description of the differentiation of the 3 germ layers in our embryoids (**Figure 1-6, Suppl Fig. 1-24**).

We performed scRNA seq (**Fig. 8-10, Supplementary Fig. 19-23**) in addition of Bulk RNA (**Fig. 7, Supplementary Fig. 17-18**) and provide an extensive analysis for identifying the structures made in our embryoids (e.g.: clusters identity, pseudotime differentiation trajectory analysis).

Quantification/details of reproducibility

For each marker, we provide the number of experiments and of biological samples we analyzed as well as the variability in gene expression we observed, with images associated with variations from the main pattern and the percentage of embryoids associated to these variations (**Supplementary Figures 3, 4, 7, 9, 10, 12-14**).

Comparison with other published work

This is addressed throughout the manuscript when appropriate and in the discussion.

Detailed responses to Referee 3

1. Statistics are insufficient throughout. For every observation, the authors need to note in the main text how many aggregates were analyzed and how many showed the expected phenotype.

To answer this request and to make the paper as easy to read as possible, we built a number of additional supplementary figures and tables associated (**Supplementary Fig. 3, 4, 7, 9, 10, 12-14**) associated to the main figures (**Fig. 1-6**) in which we describe in details the number of samples, number of experiments, statistics, and describe the variability of *in situ*

staining by reporting, with representative images, variant expression patterns observed in the labeled population as well as their frequency in the population analyzed. This analysis required a lot of space and could not fit in the main text, this is why we put it in Supplementary Figures. Nevertheless, to make sure the reader will see these important results, we mentioned these data in each legend figure ‘Number of experiments performed, biological samples analyzed and variability in gene expression is provided in associated **Supplementary Fig(s)**. In addition, when expression of a marker was not detected in the majority of the samples or when we got a low number, this is mentioned in the main text.

e.g.: ‘in about 15% of embryoids displaying a dorsal *Sox1-GFP(+)* epithelium, we observed the folding of the neural plate around D7 (**Fig. 6a**)’

We hope the strategy we undertook will be acceptable to Referee 3.

Relatedly, information about reproducibility is lacking. The variability in size and shape parameters (e.g. length, width, volume etc) needs to be presented. Similarly, for each staining pattern, variability should be noted. Presumably not all aggregates look the same and the authors should accurately convey the spectrum of phenotypes.

In addition to the description of the variability described above, we added a Supplementary figure displaying the variability in morphology of embryoids at D7 (**Supplementary Fig. 3**)

3. It would be very helpful in understanding the model to know which tissues are contributed by the pretreated cells. Similar images to Figure 1a at later time points would be informative. Is the node-like structure derived from the graft? What about the notochord-like structure? Do both the pretreated and untreated cells contribute to all three germ layers?

To identify the origin of cells (originating from the signaling center aggregate or from the naive (uninstructed) aggregate), we generated embryoids with combinations of *GFP* labeled ESC (129-*GFP* that express *GFP* constitutively) and unlabeled E14TG2a ESC. Our sets of experiments, presented in **Supplementary Fig. 15 and 16**, revealed that the neuroectodermal cells of the neural plate derive from the naive territory, visceral endoderm from both the signaling center and the naive territory, definitive endoderm and most of the mesoderm derive from the signaling center.

*4. The experiment in figure 2 requires quantification. The authors should plot the positions of *GFP* expressing cells along the axis of the embryoid.*

This is done and the curve associated with the measurement and the raw data of the position of cells along the AP axis are now presented in **Supplementary Fig. 6 and Supplementary Table 7**.

*5. In Figure 3, 5, the tube structures of the gut and neural tube are evident in the immunofluorescence but more difficult to observe in the in situ. However, the immunofluorescence images are not stained for fate markers so it is difficult to confirm the interpretation. It would be good to costain with markers of fate (e.g. Phalloidin/*Sox17* in figure 3, Phalloidin/*Pax6* in Figure 5).*

For the endoderm, we extended our old **Figure 3** in a new Figure 3 and document the expression of a collection of markers as well as we provide a Phalloidin-*sox17* staining (**Fig. 3k**) showing the expression of *Sox17* in the polarized epithelium surrounding the inner cavities.

For the ectoderm, we have now extended our description and staining to 2 figures (Fig. 5 and Fig. 6) and analysis of scRNAseq. We performed a SOX2-Phalloidin double labeling to establish the neurectodermal nature of the epithelium (**Fig. 6b**) as well as in situ for Sox2 presented in transverse section (**Fig. 6c and 6h**).

6. In figure 4h, it is difficult to see the network of vessels the authors refer to. Better data should be provided or the claims softened.

We extended Figure 4 and provide now additional panels (7 panels **Fig. 4n-t** and schematics instead of one panel in the previous version of Figure 4) to analyze *Kdr* expression pattern from D5 to D8 and to better document formation of the vasculature network.

Similar to the mouse embryo, expression of *Kdr* in embryoids (**Fig. 4n-o**) is detected in a ventral anterior territory, in tubular-like structures resembling endocardial-like tubes. At D5.5, *Kdr* expression (**Fig. 4p**) is observed with a pattern very similar to mouse early vascular development (cartoon in **Fig. 4q**), with different domains of early vascular-like development, both in the cardiac area as well as in two rods of cells aligned antero-posteriorly that resemble the growing dorsal aortae preceded by scattered aorta progenitor cells. Scattered cells are observed laterally, at the level of the heart. They may represent the precursors of the vitelline and cardinal veins, which appear at that location in the mouse embryo. At D6, *Kdr* expression pattern appears more complex in the embryoids (**Fig. 4q**), showing both a high density of scattered blood vessel progenitors and a strong labeling of a large dorsal vessel (either dorsal aorta or posterior cardinal vein) that connect perpendicular smaller vessel, reminiscent of intersomitic blood vessels of the mouse embryo. Two days after, the number of blood vessels has strongly increased while less progenitors are observed and the embryoids start to be covered by a reticular network of blood vessels (**Fig. 4r-s**) that increases in density through angiogenesis. High magnification in **Fig. 4t** shows the formation of novel capillaries likely through sprouting from pre-existing larger vessels, sprout outgrowth and sprout fusion as described for angiogenesis in the mouse embryo.

7. The data in Figure 6 don't add much to that in Figure 7,8 and could be moved to supplementary material.

We now present the bulk RNA data in one figure of the main text in which we compare transcriptomes of embryoids at D7 and D8 with transcriptomes of mouse embryo at E8.5 and E9.0 (**Fig. 7**) and moved the comparison with Controls in **Supplementary Fig. 17, 18**.

8. The statement in lines 69-71 is not entirely correct with regard to mammalian embryos. Not all signals come from the "embryo proper" but many are secreted from extraembryonic tissues. As, for example, the trophoctoderm is in contact with the edges of the cup-shaped embryo, it may indeed provide signals particularly at the periphery.

We apologize for this approximation. This is now corrected.

9. The discussion is rather biased in comparing this model to others. The authors state that this model is more "controlled", however, they do not define this word. In some ways, it is easier to control the precise concentration of an extracellular inducer than the concentrations of Nodal/Wnt which are induced in the pretreated cells, which will vary. It is also possible to withdraw extracellular inducers but the cell graft cannot be undone. The authors state that the fact that asymmetry arises without experimental control in other models is a drawback,

however, I consider this a feature – symmetry breaking is one of the most important and interesting processes in mammalian development and the authors have bypassed this by experimentally breaking the symmetry. Overall, the authors should be more clear and precise about the advantages and disadvantages of their model compared to others.

We apologize for being unclear and not precise enough in presenting our model.

‘Control’ and its advantages

We create a local signaling center with a morphogen (BMP4) that in response induces the expression of morphogens NODAL and WNT. In this first step, we control the instruction (BMP dosage, number of cells, timing of instruction, duration of instruction). Our goal is to avoid self-organization of the aggregates.

One important point is that *in vivo* the developmental programs that control patterning and morphogenesis of the embryo are not induced by soluble signals present in the liquid medium surrounding the embryo.

Induction of the developmental programs in gastruloids or TLS results from incubation of ESC aggregates in a culture medium containing a small molecule agonist of the WNT/beta catenin signaling pathway, which is homogeneously distributed around the ESC aggregates and stimulates this structure isotropically. All superficial cell aggregates are exposed to the same level of stimulation, which depends on the concentration of the WNT agonist. While the initial spherical symmetry occurs in these entities, it is likely the result of induction of mesoderm in superficial cells (that may be stimulated more by the WNT agonist) and a cell sorting of mesodermal and ectodermal cells resulting in the formation of a *Bra* expressing group of cells that ‘breaks’ the symmetry and define a posterior territory. However, during the process of activation of the WNT/beta catenin signaling pathway, all cells of the aggregates are likely to have been stimulated by the WNT agonist CHIR (with probably stronger stimulation for cells at the surface of the aggregate and less activation of the WNT pathway in deep cells). Because all cells were stimulated to activate the downstream WNT/beta catenin signaling pathway, cells of the anterior neural domain, that require the absence of WNT signaling to form brain tissues are stimulated by WNT agonist. This may be why this results in a posteriorization of the gastruloid/TLS that are then restricted to post-occipital fates.

Instructing with an asymmetrically located morphogen that initiates the start of the developmental programs and downstream response factors allows the succession of developmental steps to take place, similar the mouse embryonic development. We therefore make all three germ layers that are organized relative to each other as it is in vertebrates, around an axial mesodermal notochord-like domain (the landmark of the phylum Chordata) with a neural plate/tube above this axial structure and the endoderm below. This organization is observed from anterior to posterior allowing signals (such as SHH to be released from the axial midline, inducing ventral neuronal fate as shown for *Olig2*).

General advantages of our embryoid model

The method we use brings a number of advantages such as the possibility of changing various parameters:

We can manipulate the size of the signaling center, we can modulate the level of activation playing with BMP4 concentration. We can mix different cell lines generating mosaic embryoids and we can design additional signaling centers (such as an anterior center secreting

WNT antagonist to protect the anterior most part of the embryoid from posteriorizing activity and allow more anterior development, a project we will start to explore soon)

Even this was not the scope of the present study, we illustrate this point in **Supplementary Fig. 24** showing that increasing BMP4 concentration produce embryoids with an increase in frequency of the formation of heart, erythrocytes and definitive endoderm/gut epithelium. For heart and endoderm, increase in BMP4 concentration also increases the size of the heart and of the gut tube.

Such a system can probably be improved, in particular by allowing embryoids to grow in media containing Matrigel and we anticipate that, similar to what has been described for gastruloids and TSL, the morphology of embryoids at late stages will be improved.

Altogether, the embryoid model we present in this study will benefit future studies of understanding and manipulating gradients, modeling diseases *in vitro*, drug screening and develop a human counterpart.

Disadvantages of our embryoid model

So far, this is a manual method and this is time consuming. However, we can make 700-1000 embryoids per week and per person.

10. The authors should revise claims about limited morphogenesis in gastruloids and priority claims about neural tube formation in vitro in light of recent publications and preprints – van den Brink et al Nature 2020, Veenliet et al bioRxiv 2020.

The study performed by Van den Brink *et al* focusses on the formation of paraxial mesoderm and somitogenesis in the gastruloids. A scRNA seq revealed that '*neuromesodermal progenitors are located in the most-posterior part of the gastruloids (cluster 7) and that more differentiated neural cells are found slightly more anteriorly*'. However, there is no *in situ*/immunolabeling presented in this study for neural plate/neural tube for AP and DV patterning, no transverse section nor characterization of the neural epithelium, (e.g.: apical-basal polarity, mitosis). As well, there is almost no mention of the presence of endoderm except for scRNAseq and there is no characterization (*in situ*/immunolabeling) of a primitive gut formation and differentiation in the gastruloids. Therefore, this is difficult for us to assess how much of the ectodermal and endodermal layers are formed and/or differentiate in gastruloids and TLS. Finally the chordal mesoderm does not seem to be present in gastruloids and TLS models, which is a strong limitation for the production of an *in vitro* model of a vertebrate embryo.

'There is much more staining necessary to claim true patterning. It is one thing to have the transcriptome show that many of the genes that are patterned are in the structures, but it is entirely a different thing to see if the cells expressing those genes are in the correct place'.

This is why in our revised version of the manuscript, we used markers as specific as possible to identify unambiguously the structures we see. We now describe (D3.5-D9) with a collection of specific markers (40 markers for *in situ* and/or immunolabeling) a much more precise description of our embryoids that makes the point that we get 3 germ layers formation and differentiation (**Figure 1-6**).

11. *The authors may wish to discuss Cederquist et al Nature Biotech 2019 which takes a similar approach to patterning in the nervous system.*

The study by Cederquist et al Nature Biotech 2019 uses human brain organoids as a model system. To correct the lack of a reproducible topographic organization their hypothesis is that introduction of a signaling center into forebrain organoids may specify the positional identity of neural tissue. They present a system to trigger SHH protein gradient in developing forebrain organoids. The result is that major forebrain subdivisions are positioned with *in vivo*-like topography. We mention this study in the discussion section.

Our embryoid system don't have a forebrain. Our hypothesis is that the initial BMP4 instruction is too potent leading to strong activity of WNT/NODAL. This is the focus of a new project intended to counteract NODAL and WNT gradient. We will engineer in a next study an anterior visceral endoderm-like center from which WNT/NODAL antagonist will be released in order to limit the anterior extent of their posteriorizing activity and to promote formation of anterior brain (also mentioned in the discussion section).

Reviewers' Comments:

Reviewer #1:

Remarks to the Author:

The authors have made great effort to address most of my comments. The quality of this manuscript (including its writing) is much improved, compared to the last version. I think this manuscript will become a reference to cite for the field of stem cell-based embryo models. Nonetheless, I still have a few comments here that I hope the authors should address, to improve accuracy and further strength the impact of this work.

1. Title. The authors should change the title to something like "Construction of a Mammalian Embryo Model from Stem Cells Organized by a Morphogen Signalling Centre".
2. In Introduction, the authors discussed 3D neural induction culture to generate neuroepithelial cysts (Ref. 16; page 69). I think the authors should add two more references here (Meinhardt, A., Eberle, D., Tazaki, A., Ranga, A., Niesche, M., Wilsch-Bräuninger, M., Stec, A., Schackert, G., Lutolf, M. & Tanaka, Elly M. 3D reconstitution of the patterned neural tube from embryonic stem cells. *Stem Cell Reports* 3, 987-999 (2014).; Zheng, Y., Xue, X., Irizarry, A. M. R., Li, Z., Shao, Y., Zheng, Y., Zhao, G. & Fu, J. Dorsal-ventral patterned neural cyst from human pluripotent stem cells in a neurogenic niche. *Science Advances* 5, eaax5933 (2019).)
3. The pseudotime differentiation analyses in Fig. 9 and Fig. 10 are interesting. Nonetheless, overinterpretation of such data analyses, without real experimental data support, can be dangerous. For example, in their Fig. 10a-e, their data somehow suggest that FB will develop later than NMP and SC, and their pseudotime plot in Fig. 8e further suggests that FB is derived from NMP, which is not true.
4. The D-V patterning data for the neural tube in Fig. 10f-v based on the scRNA-seq data are quite neat. Nonetheless, I am not completely satisfied, since the authors didn't show any specific staining data to confirm D-V patterning of the neural tube. The authors should provide real convincing experimental data to show proper D-V patterning of the neural tube in their model.

Reviewer #3:

Remarks to the Author:

The authors have responded very thoroughly to comments from the previous round of review, including a large amount of new data. I am satisfied by their responses and recommend publication. I particularly appreciate the presentation of the variety of phenotypes with quantification in the supplementary information. This is an excellent way to show this data and a good model for the field which is generally lacking a rigorous description of variability.

I have only a few minor comments:

1. There are still many grammatical inaccuracies, I would recommend careful proofreading before publication.
2. The reference to "not shown" in line 176 should be removed and the data shown.
3. The claims in lines 290-293 regarding the interpretation of the lack of Sox17 Bra coexpression are overstated. The lack of coexpression does not necessarily indicate a different origin to these cells but only that endodermal cells downregulate Bra before upregulating Sox17

ANSWERS TO REVIEWERS' COMMENTS

Reviewer #1 (Remarks to the Author):

The authors have made great effort to address most of my comments. The quality of this manuscript (including its writing) is much improved, compared to the last version. I think this manuscript will become a reference to cite for the field of stem cell-based embryo models. Nonetheless, I still have a few comments here that I hope the authors should address, to improve accuracy and further strength the impact of this work.

1. Title. The authors should change the title to something like "Construction of a Mammalian Embryo Model from Stem Cells Organized by a Morphogen Signalling Centre".

We have changed the title according to Referee's suggestion.

*2. In Introduction, the authors discussed 3D neural induction culture to generate neuroepithelial cysts (Ref. 16; page 69). I think the authors should add two more references here (Meinhardt, A., Eberle, D., Tazaki, A., Ranga, A., Niesche, M., Wilsch-Bräuninger, M., Stec, A., Schackert, G., Lutolf, M. & Tanaka, Elly M. 3D reconstitution of the patterned neural tube from embryonic stem cells. *Stem Cell Reports* 3, 987-999 (2014).; Zheng, Y., Xue, X., Irizarry, A. M. R., Li, Z., Shao, Y., Zheng, Y., Zhao, G. & Fu, J. Dorsal-ventral patterned neural cyst from human pluripotent stem cells in a neurogenic niche. *Science Advances* 5, eaax5933 (2019).)*

These two references have been added to the references of the manuscript (ref #18 and 19).

3. The pseudotime differentiation analyses in Fig. 9 and Fig. 10 are interesting. Nonetheless, overinterpretation of such data analyses, without real experimental data support, can be dangerous. For example, in their Fig. 10a-e, their data somehow suggest that FB will develop later than NMP and SC, and their pseudotime plot in Fig. 8e further suggests that FB is derived from NMP, which is not true.

Our data do not suggest that the forebrain will develop latter and NMP and Spinal Cord: The high pseudotime value indicates that anterior neuroectoderm differentiates first and is well differentiated at D8 while at that stage NMP cells are only at the beginning of their differentiation (low pseudotime value). We wrote again this paragraph to make sure that it is unambiguous and that it would suggest that the anterior neuroectoderm is derived from the NMP. We did the same for the analysis of mesodermal pseudotime.

4. The D-V patterning data for the neural tube in Fig. 10f-v based on the scRNA-seq data are quite neat. Nonetheless, I am not completely satisfied, since the authors didn't show any specific staining data to confirm D-V patterning of the neural tube. The authors should provide real convincing experimental data to show proper D-V patterning of the neural tube in their model.

The presence of a DV axis for the neurectoderm is shown with expression of *Shh* and *Olig2* in the ventral neural tube and the presence of neural crest cells expressing *Sox10* at the border of the dorsal side of this neural tube. Further evidence are provided in the scRNA-seq data that

allowed us to detect at D8 the expression of markers for the different domains of the neural tube along the DV axis with a higher sensitivity than in situ or immunolabelling at this stage.

Reviewer #3 (Remarks to the Author):

The authors have responded very thoroughly to comments from the previous round of review, including a large amount of new data. I am satisfied by their responses and recommend publication. I particularly appreciate the presentation of the variety of phenotypes with quantification in the supplementary information. This is an excellent way to show this data and a good model for the field which is generally lacking a rigorous description of variability.

I have only a few minor comments:

1. There are still many grammatical inaccuracies, I would recommend careful proofreading before publication.

The manuscript has been read and edited for grammar and style by a native English speaker, Dr. R. Bloodgood.

2. The reference to “not shown” in line 176 should be removed and the data shown.

Data are now shown in Supplementary Figure 5.

3. The claims in lines 290-293 regarding the interpretation of the lack of Sox17 Bra coexpression are overstated. The lack of coexpression does not necessarily indicate a different origin to these cells but only that endodermal cells downregulate Bra before upregulating Sox17

We changed the wording of the statement to: “We didn’t observe any cells expressing both endodermal and mesodermal markers. This supports, in embryoids, the published observation that, in both the mouse embryo and in gastruloids, cells may already be committed at the time of gastrulation to endodermal or mesodermal fates rather than being bipotential, and representing a mesendodermal population.”

In the manuscript, changes in the text in response to Reviewer requests are in **green**. The abstract has been modified in response to editor request. The change in the text for grammar and English mistakes are indicated in **red** for deleted text and **in blue** for additions or changes.